# „Quantification of Drainable Water Storage Volumes on Landmasses and in River Networks based on GRACE and River Runoff using a Cascaded Storage Approach – First Application on Amazon"

Johannes Riegger [1]

[1] Institute for Modelling Hydraulic and Environmental Systems, University of Stuttgart, Germany

*Correspondence to*: Johannes Riegger (Johannes.Riegger@iws.uni-stuttgart.de)

**Abstract.**

The knowledge of water storage volumes in catchments and in river networks leading to river discharge is essential for the description of river ecology, the prediction of floods and specifically for a sustainable management of water resources in the
context of climate change. Measurements of mass variations by the GRACE gravity satellite or by ground based observations of river or groundwater level variations do not permit the determination of the respective storage volumes, which could be considerably bigger than the mass variations themselves.

For fully humid tropical conditions like the Amazon the relationship between GRACE and river discharge is linear with a
phase shift. This permits to determine the hydraulic time constant and thus to quantify the total drainable storage directly from observed runoff, if the phase shift can be interpreted as the river time lag. As a time lag can be described by a storage cascade, a lumped conceptual model with cascaded storages for the catchment and river network is set up here with individual hydraulic time constants and mathematically solved by piecewise analytical solutions.

Tests of the scheme with synthetic recharge time series show that a parameter optimization either versus mass anomalies or runoff reproduces the time constants for both, the catchment and the river network $\tau_C$ and $\tau_R$ in a unique way, and hence permits an individual quantification of the respective storage volumes. The application to the full Amazon basin leads to a very good fitting performance for total mass, river runoff and their phasing (Nash-Sutcliffe for signals 0.96, for monthly residuals 0.72). The calculated river network mass highly correlates (0.96 for signals, 0.76 for monthly residuals) with the
observed flood area from GIEMS and corresponds to observed flood volumes.

The fitting performance versus GRACE permits to determine river runoff and Drainable storage volumes from recharge and GRACE exclusively i.e. even for ungauged catchments. An adjustment of the hydraulic time constants ($\tau_C$, $\tau_R$) on a training period facilitates a simple determination of Drainable storage volumes for other times directly from measured river discharge and/or GRACE and thus a closure of data gaps without the necessity of further model runs.

# 1 Introduction

In the context of water resources management and climate change there is an ongoing discussion on how to assess available water resources i.e. the storage volumes which can be used for water supply in a dynamic way beyond the limitations of sustainable extraction rates. The maximum average extraction rate for a sustainable use of water resources is limited by the long-term recharge of a catchment Sophocleous (1997); Bredehoeft (1997), however, this rate based definition of groundwater stress only allows an assessment of water resources with respect to long-term sustainability and does not permit short term management in order to satisfy specific water demands. Thus the knowledge of water resources involved in the water cycle contributing to river discharge such as parts of the groundwater or surface water system is essential.

Very little attention has so far been given to the quantification of the storage volumes of renewable water resources participating in the dynamic water cycle driven by precipitation P, actual evapotranspiration $ET_a$ and river runoff R. The reason for this is seen in the problem that observations of time variant groundwater or river levels only permit the estimation of volume changes yet no absolute storage volumes, which could be considerably bigger.

Natural systems consist of many different storage components like canopy, snow/ice, surface, soil, unsaturated/saturated underground, drainage system etc. Direct measurements of storage volumes from water or pressure levels are problematic as they are based on assumptions and approximations. They are based on point measurements and quite rare on large spatial scales compared to the heterogeneity scale of the respective compartments. This leads to large interpolation errors. In addition, the storage coefficients for porous media describing the relationship between the measurable groundwater heads or capillary pressure on the one hand, and storage volume or absolute soil saturation on the other hand, are insufficiently known on large scales. Remote sensing data have been limited to near surface water storage (open water bodies, soil) up until now and are thus of limited benefit for the quantification of water storage with respect to accuracy and coverage due to methodological constraints Schlesinger(2007).

In contrast to discharge-less basins and/or arid areas, which are nearly exclusively driven by precipitation and evapotranspiration, the storage dynamics of catchments draining into a river system allows to address the hydraulically coupled storage compartments via their contributions to river discharge. These comprise groundwater, surface water, the river network and temporarily inundated areas. All storages draining into the river system by gravity are referred to as "Drainable" storage here. So, aquifers or parts of them not draining into the river system without an energy input are not considered here.

River Runoff  R(t) = Q(t)/A (corresponding to river discharge Q(t) from the related catchment area A) is driven by the storage height or mass density $M_{Storage} = V_{tot}/A$ of all superposed hydraulically coupled storages and is determined by their runoff storage (R-S) relationship. For time periods with no recharge or losses of water (as by ETa) i.e. with no processes

affecting $M_{Storage}$ other than by river discharge Q, a linear runoff storage (R-S) relationship $R(M) = M/\tau$ leads to an exponential decrease in river discharge or streamflow Q(t) depending on the related hydraulic time constant $\tau$:

$$Q(t) = Q(t_0) \cdot e^{-\frac{t-t_0}{\tau}} \tag{1}$$

For this case the corresponding total "Drainable" Storage in terms of mass density $M_{Storage}$ at any given time $t_0$ can be determined by an infinite temporal integration over river discharge Q(t) from the corresponding catchment area A starting at time $t_0$ :

$$M_{storage}(t_0) = \frac{V_{tot}(t_0)}{A} = \frac{1}{A} \cdot \int_{t_0}^{\infty} Q(t)dt = \frac{Q(t_0)}{A} \cdot \int_{t_0}^{\infty} e^{-\frac{t-t_0}{\tau}} dt = \tau \cdot \frac{Q(t_0)}{A} = \tau \cdot R(t_0) \tag{2}$$

Contributions from several storage compartments (with individual time constants) superpose, if they drain in parallel and if there is no feedback from the river system. For this case, there is a wide range of time series analysis methods Tallaksen (1995), which allow to separate the flow components into fast, medium or slow and the corresponding surface, interflow or groundwater flow contributions according to their individual time constants. Thus, measurements of the different time constants allow to determine the Drainable Storage of the respective storage compartment and the corresponding mean

Drainable Storage :

$$\overline{M}^X = \overline{R} \cdot \tau_X = \overline{N} \cdot \tau_X \ . \tag{3}$$

from mean runoff R or recharge N.

    On global scales the absolute storage volume of the Drainable Storages can be determined from runoff time series directly, if

there are distinct and long enough periods of negligible or even negative recharge (actual evapotranspiration $ET_a >$ precipitation) as it occurs in seasonally dry regions (Niger, Mekong, some Amazon sub-catchments etc.). From the purely exponential decrease in river discharge the time constant can be determined directly from a curve fit as shown in Fig.1b for Amazon sub-catchments. If the dry period is long enough the sequence of different time constants taken from the discharge curve even permits a discrimination between the fast response by overland flow and the slow response by the groundwater

system.

    Catchments with permanent input i.e. no periods of negligible recharge, however, do not show an exponential behaviour for discharge. For these cases the hydraulic time constant cannot be taken from discharge dynamics directly, but has to be estimated by hydrological models These intend to describe the large number of storages distributed over the catchment by

the assumed processes and calibrate the involved parameters by their respective superposed flows versus the observed river

discharge. The main difficulties in verifying large or Global Scale Hydrological Models or Land Surface Models (GHMs / LSMs) consist in the quantification of local individual storage volumes and related flows by local groundbased measurements. Thus, even though distributed hydrological models very much support an understanding of processes in the water cycle the limitation of the calibration versus river discharge exclusively introduces an ambiguity in the impact of contributing processes and the related storages and flows.

Since 2013 GRACE observations of the time-variable gravity field provide monthly distributions of mass density on large spatial scales $>\sim 200000 km^2$ Tapley et al. (2004). However, as the water storage in different compartments (snow, ice, vegetation, soil, surface-, ground- water etc.) superposes with all other terrestrial (geophysical) masses, only the time variant part of the GRACE signal can be used to quantify the Terrestrial Water storage (TWS) anomalies (monthly mass signals minus long term average), but not the related absolute storage volumes. Nevertheless, this for the first time permits a direct comparison of measured TWS and observed river runoff $R_o$. Surprisingly some global hydrological models (GHMs) showed a considerable phase shift between measured mass anomalies by GRACE and river discharge as well as between calculated and measured runoff and an underestimation of mass signal amplitudes Güntner et al. (2007); Chen et al. (2007); Schmidt et al. (2008); Werth et al. (2009); Werth et al. (2010) even though they comprise a large number of storages like soil, surface water, groundwater etc. This is emphasized by Scanlon et al. (2019), who for tropical basins recognize the main cause of the discrepancies in insufficient storage capacity and lack of surface water inundation.

The direct comparison of GRACE anomalies and river runoff on large spatial and monthly time scales by Riegger and Tourian (2014) revealed that measured runoff- storage (R-S) diagrams show hysteresis curves of distinct form and extent (Fig1a,b), which are characteristic for different climatic conditions (like fully humid, seasonally dry or boreal) and can be explained considering recharge and runoff properties (Fig1c,d) .

Thus for example, catchments in fully humid conditions (like the full Amazon basin upstream Obidos (295) and some of its catchments like upstream Manacapuru (501)) with a permanent input i.e. only positive recharge (Fig.1c) show a counterclockwise hysteresis (Fig.1a). If this can be fully described by a positive phase shift river runoff and storage behave like a Linear Time Invariant (LTI) system Riegger and Tourian (2014) i.e. the R-S relationship is linear, if the phase shift is adapted as shown in Fig.1a. For this case the hysteresis can be purely assigned to a time lag. Once the phase shift is adapted the slope in the R-S diagram corresponds to the hydraulic time constant via $\tau = M/R$. The time constant and the reasonable assumption of a proportional R-S relationship (no runoff for empty storage) then facilitates the quantification of the Drainable Storage, (Eq. 3), i.e. the volume related to the hydraulically coupled storage compartment, which drains by gravity.

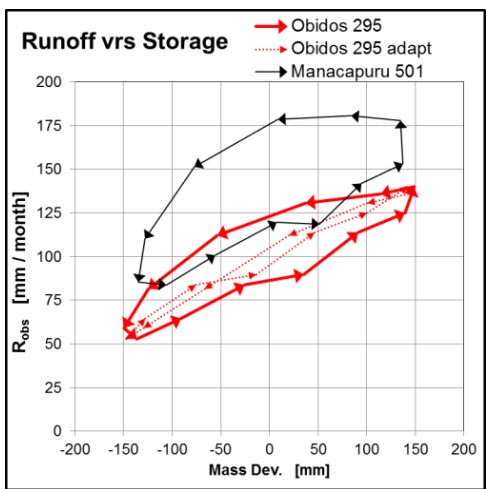

Fig.1a: R-S diagram with counter clockwise hysteresis for mean monthly observed runoff $R_o$ versus GRACE dM for fully humid catchments incl. a phase adaption for Amazon upstream Obidos Riegger and Tourian (2014)

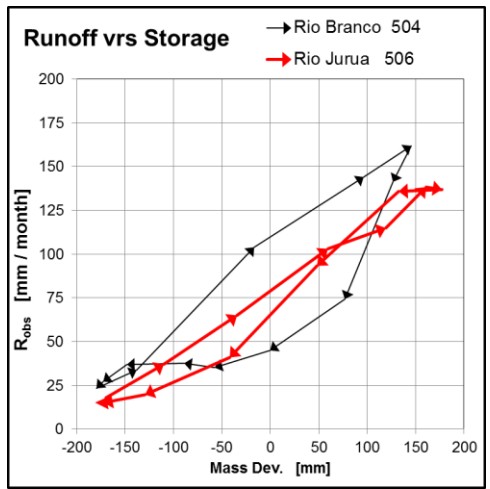

Fig.1b   R-S diagram with clockwise hysteresis for mean monthly observed runoff $R_o$ versus GRACE anomaly dM for seasonally dry catchments in the Amazon basin (Riegger and Tourian (2014)

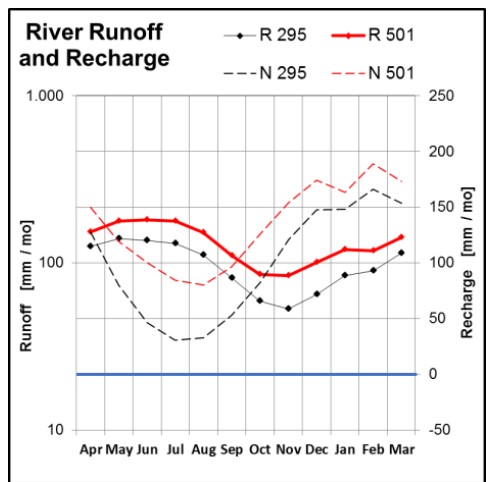

Fig.1c: Mean monthly Runoff R and Recharge N for fully humid catchments in the Amazon basin (Log scale for R)

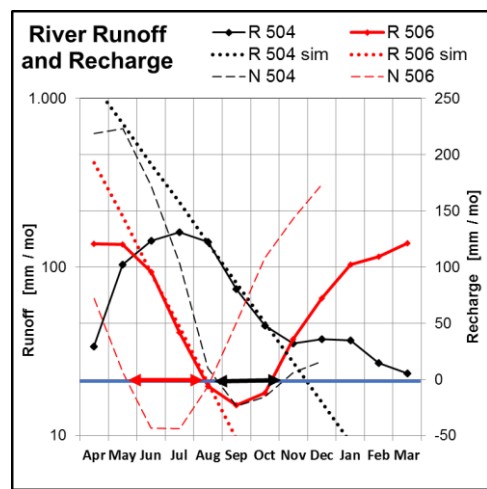

Fig.1d: Mean monthly Runoff R and Recharge N for seasonally dry catchments in the Amazon basin (Log scale for R) incl. exponential fittings for Runoff $R_{sim}$

In contrast, catchments with distinct periods of zero or negative recharge (like Niger, Mekong or Rio Branco (504), Rio Jurua (506) in the Amazon basin (Fig.1b)) show a clockwise hysteresis in the R-S diagram and a form, which is determined by an increase in mass and runoff during wet periods, a decrease in mass and runoff with different slopes corresponding to different time constants and a possible mass loss without a related runoff (by negative recharge (Fig.1d) by

evapotranspiraton from the soil zone) during dry periods. This type of hysteresis is determined by storage changes not connected with river discharge (uncoupled) and cannot be explained by a time lag as it is not causal.

The consequence from the above discussion is that the determination of the hydraulic time constant and thus the Drainable storage is only possible for catchments for which the hysteresis is fully explained by a positive phase shift i.e. uncoupled storages are either negligible or can be separated from GRACE mass by other means (as shown below for boreal regions).

Based on this method Tourian et al. (2018), apply an adaption of the phase shift using a Hilbert transform in order to determine the hydraulic time constants and the total Drainable water storage for the sub catchments of the Amazon basin without a consideration of the form of the R-S hysteresis. To be sure, this leads to reasonable results for the sub catchments with permanent input (Fig.1 a, c) for which the time dependent uncoupled storage is negligible. However, for Rio Branco (504) or Rio Jurua (506) this condition is not fulfilled as the hysteresis is determined by mass changes in the uncoupled storage and by runoff with different time constants (Fig.1b, d). For these catchments the exclusive adjustment of the phase shift leads to negative time lags - which are not physical - and as a consequence to misleading time constants and thus to considerable errors in the determination of Drainable storage volumes.

The accurate description of the R-S hysteresis of a catchment and its river network is the prerequisite for an accurate description of the system dynamics and the related storage volumes on the land masses (canopy, soil, overland flow, saturated / unsaturated underground) and in the river network.

Recent developments in river routing schemes of global hydrologic models with a hydrodynamic modelling of the flow in the river network system have successfully dealt with the description of phase shifts generated by the time lag in the river network Paiva et al. (2013); Luo et al. (2017); Siqueira et al. (2018); Getirana et al. (2017a) emphasize the importance of integrating an adequate river routing schemes not only for an improved phase agreement with observed river discharge but also for an appropriate fit of the total mass amplitude to GRACE by the inclusion of the corresponding river network storage. Yet a hydrodynamic modelling of a complete river network system for the determination of the river network time lag and storage means a huge modelling effort Getirana et al. (2017b).

A far more simple approach is presented by Riegger and Tourian (2014), describing the system by macroscopic variables summarizing all coupled storage compartments on landmasses and in the river network and analogously all uncoupled storage compartments in one respective single storage by their effect on the R-S relationship. The intention of such a "top-down" or lumped approach is to integrate the catchment scale water balance and describe the system by large scale variables and parameters, which are directly measurable or adjustable. For this purpose recharge based on moisture flux divergence or catchment water balance using GRACE can be used which are quite accurate, yet limited to global scales (see below). Thus, opposite to distributed hydrological models which are based on spatially / temporally distributed data (for hydrometeorological input, local storage conditions in vegetation, soil and underground) and a detailed description of

internal processes - which cannot be verified locally at present – this "top-down" approach uses measured catchment scale input, storages and runoff. Where necessary and possible catchment scale parameters are used to separate coupled and uncoupled storages (like MODIS snow coverage for boreal regions, Riegger and Tourian (2014)). In addition the time lag between storage and river discharge need not be explicitly described by an excessive routing scheme. Instead the related phase shift can be adapted by mathematical methods. This leads to a description of the system behaviour with high accuracy (Nash Sutcliffe 0.97 for full Amazon) by an adaption of only two parameters, the hydraulic time scale and the phase shift, even though the physical cause of the phase shift is not addressed explicitly.

A disadvantage of the above approaches Riegger and Tourian (2014); Tourian et al. (2018) is that it does not permit to quantify the individual Drainable storage volumes on landmasses and in the river network separately, but only the total Drainable volume of the catchment. The information contained in the phase shift or time lag is not used for a quantification of the river network storage volume. Yet, as observations of inundated areas in river networks such as from the GIEMS "Global Inundation Extent from Multi-Satellites" project Prigent et al. (2007); Papa et al. (2008); Papa et al. (2013) and hydrodynamic models of the river network Paiva et al. (2013), Getirana et al. (2017b); Siqueira et al. (2018) indicate a considerable contribution of river network storage corresponding to a non negligible time lag, the river network storage must be considered in the integration of the total catchment water balance. As a sequence of storages (cascaded storages) leads to a time lag i.e. a phase shift Nash (1957) and storages draining in parallel (as for overland and groundwater flow) just lead to a superposition (with no time lag), a storage cascade is considered as an appropriate description to account for a time lag.

This paper explores the accuracy and uniqueness of a lumped, top down approach called "Cascaded storage approach" based on the integration of given recharge in the water balance utilizing a cascade of a catchment storage and a river network storage for a simple description of the observed time lag and the individual storage volumes. This permits to describe the system with a minimum number of macroscopic observation data and an adaption of only two parameters, the hydraulic time constants of the catchment and the river network. These time constants then could be used for nowcasts or even forecasts (within the time lag) of river discharge and/or Drainable storage volumes directly from measurements without the need for further modelling.

The paper is structured as follows: Section 2 presents the mathematical framework of piecewise analytical solutions of the water balance equation for a cascade of catchment and river network storages. It also contains the description of observables, which permit the comparison of calculated and measured values. The Single Storage approach is handled as the specific case for a negligible river network time constant. In section 3, the properties of the Cascaded Storage approach and its impact on the performance of the parameter optimization are described for synthetic recharge data and compared to the "Single Storage" approach. Based on the Cascaded Storage approach a fully data driven approach is presented which permits a simplified determination of the Drainable storage volumes directly from measurements without the need of further model

runs. In section 4 the approach is applied to data from the Amazon basin and evaluated versus measurements of GRACE mass, river runoff and flood area from GIEMS. The results are compared to GHM / LSM studies. In section 5 the approach and its performance and limitation is discussed. Possible future investigations in order to overcome some of its limitations are sketched. Conclusions are drawn in section 6

## 2 Mathematical framework

In order to investigate the impact of a non negligible river water storage on the time lag in the river system, the water balance of the total system comprising both the catchment and river network storage has to be considered. A conceptual model corresponding to a Nash-cascade Nash (1957), called "Cascaded Storage" approach here, is set up with individual time constants for the different storages and with the following properties:

- Surface water and shallow groundwater storages on the land mass which are draining into the river network and are being fed by recharge are summarized to a so called "Catchment" storage $M^C$ with time constant $\tau_C$. Overland and groundwater flow from the land masses are summarized to a "Catchment" runoff $R^R$.

- River runoff (river discharge / catchment area) which addresses hydraulically the flow in the river channel network including inundated areas is determined by its hydraulic time constant $\tau_R$. The respective river network storage $M^R$ is assumed to be instantaneously distributed within the river network system. Internal routing effects, which might lead to an additional delay in streamflow response, are not considered.

- Any possible hydraulic feedback from the river to the catchment system is assumed to be negligible.

- Temporal variations of uncoupled storage compartments like soil or open water bodies are considered as negligible.

These conditions are chosen for the sake of conceptual and mathematical simplicity It has to be emphasized here that for a general applicability on a global coverage several coupled storages with different time constants and different uncoupled storage compartments with their respective time dependency have to be considered of course. For fully tropical climatic conditions with permanent recharge however (as for the full Amazon basin) variations in the soil water storage are negligible and the different dynamics of overland and groundwater flow cannot be distinguished. Thus, applications of this first approach are limited to catchments for which the hysteresis can be fully described by a time lag, i.e. no impacts of other coupled or uncoupled storages exist.

The following abbreviations are used in the mathematical description (Table.1):

| Abbreviation | Description | Units: general / for application |
|---|---|---|
| N | recharge = (precipitation - actual evapotranspiration) | volume area$^{-1}$ time$^{-1}$ [mm month$^{-1}$] |
| $M^C$ | Storage mass catchment | mass density in equivalent water height [mm] |
| $\tau_C$ | Time constant catchment | time unit [month] |
| $R^C$ | Runoff catchment | volume area$^{-1}$ time$^{-1}$ [mm month$^{-1}$] |
| $\omega_{MC}$ | Phasing catchment mass | time unit [month] |
| $M^R$ | Storage mass river network | mass density in equivalent water height [mm] |
| $\tau_R$ | Time constant river network | time unit [month] |
| $R^R$ | Runoff river network | volume area$^{-1}$ time$^{-1}$ [mm month$^{-1}$] |
| $\omega_{RR}$ | Phasing river network mass | time unit [month] |
| $M^T$ | Storage mass total system | mass density in equivalent water height [mm] |
| $\tau_T$ | Time constant total system | time unit [month] |
| $\omega_{MT}$ | Phasing mass total system | time unit [month] |
| Ro | Observed river runoff | volume area$^{-1}$ time$^{-1}$ [mm month$^{-1}$] |
| GRACE | GRACE mass anomaly | mass density in equivalent water height [mm] |
| GIEMS | Flood area | area [km$^2$] |
| Prefix "d" | indicates signal anomalies from long term mean (anomalies) | |
| Suffix "m" | indicates mean values on the intervals | |

Table.1: Abbreviations in the mathematical descriptions:

The total system behaviour is described by two balance equations, one for catchment storage (Eq. 4) and one for river storage (Eq. 6)::

    1. catchment storage

$$\frac{\partial}{\partial t} M^C(t) = N(t) - R^C(t) = N(t) - \frac{1}{\tau_C} \cdot M^C(t) \qquad (4)$$

10          with

$$R^C(t) = \frac{1}{\tau_C} \cdot M^C(t) \tag{5}$$

2. river storage

$$\frac{\partial}{\partial t} M^R(t) = R^C(t) - R^R(t) = R^C(t) - \frac{1}{\tau_R} \cdot M^R(t) \tag{6}$$

with

$$R^R(t) = \frac{1}{\tau_R} \cdot M^R(t) \tag{7}$$

with a proportional R-S relationship for hydraulically coupled storages. N denotes the recharge as input, $R^C$ the catchment runoff from the catchment storage $M^C$, which cannot be measured directly on large spatial scales, and $R^R$ the river runoff from the river network storage $M^R$ which can be measured at discharge gauging stations.

The water balance equation, Eq.(4), for the catchment is generally solved by:

$$M^C(t - t_0) = M^C(t_0) \cdot e^{-\frac{t - t_0}{\tau_C}} + \int_{t_0}^{t} N(w) \cdot e^{\frac{w - t}{\tau_C}} \cdot dw \tag{8}$$

where $M^C(t_0)$ is the initial condition and $N(t)$ the time dependent recharge.

For recharge $N(t)$ being given with a certain temporal resolution in time units or by periods of piecewise constant values and arbitrary length (stress periods) the recharge time series can be described as :

$$N(t) = \sum_{i=0}^{n-1} N_{i+1} \cdot \gamma_{i+1}(t) \quad \text{with} \quad \gamma_{i+1}(t) = \begin{cases} 1 \\ 0 \end{cases} \quad for \quad \begin{cases} t \in [t_i, t_{i+1}] \\ t \notin [t, t_{1+1}] \end{cases} \quad \text{for each interval } [t_i, t_{i+1}] \tag{9}$$

For calculation convenience Eq. (8) can be solved successively for each stress period using the values at the end of the last period as starting value, which leads to the piecewise analytical solution for catchment mass for a time $t \in [t_i, t_{i+1}]$ in stress period $_{i+1}$ :

$$M_{i+1}^C(t - t_i) = M_i^C(t_i) \cdot e^{-\frac{t - t_i}{\tau_C}} + N_{i+1} \cdot \tau_C \cdot \left( 1 - e^{-\frac{t - t_i}{\tau_C}} \right) : \tag{10}$$

The respective catchment runoff $R^C$ based on Eq. (5) and $M^{CC}$ from Eq. (10) is used as input for the river network water balance, Eq.(6), and leads to the general solution for the river network storage $M^R$ :

$$M^R(t - t_0) = M^R(t_0) \cdot e^{-\frac{t-t_0}{\tau_R}} + \int_{t_0}^{t} R^C(u) \cdot e^{\frac{u-t}{\tau_R}} \cdot du \tag{11}$$

and the iterative solutions for time $t \in \lfloor t_i, t_{i+1} \rfloor$ in stress period $_{i+1}$ :

$$M_{i+1}^R(t - t_i) = M_i^R(t_i) \cdot e^{-\frac{t-t_i}{\tau_R}} + N_{i+1} \cdot \tau_R \cdot \left(1 - e^{-\frac{t-t_i}{\tau_R}}\right) + \left[M_i^C(t_i) - N_{i+1} \cdot \tau_C\right] \cdot \frac{\tau_R}{\tau_C - \tau_R} \cdot \left(e^{-\frac{t-t_i}{\tau_C}} - e^{-\frac{t-t_i}{\tau_R}}\right) \tag{12}$$

The total mass $M^T$ is then given by :    $M_i^T = M_i^C + M_i^R$. $\tag{13}$

The mixed term in Eq. (12) and thus the total mass are commutative in ($\tau_C$, $\tau_R$) and show a singularity at $\tau_C = \tau_R$ with an asymptotic value. For $\tau_R > \tau_C$ solutions also exist with analogous values in total mass $M^T$ for $M^R > M^C$.

It has to be emphasized here, that the piecewise analytical solutions for time periods of constant recharge provide a
mathematical solution for an arbitrary temporal resolution without numerical limitations. Finite Difference solutions are limited by stability criteria      $(t_{i+1}-t_i)<\tau$ and accuracy criteria $(t_{i+1}-t_i)<\tau/10$ for the smallest $\tau$. Analytical solutions facilitate an exact calculation of the response of the river network during the time interval of constant recharge (though the time constant of the river network could be much shorter than the time interval or the time constant of the catchment). Thus the very high temporal discretization, which otherwise would be needed using a Finite Difference scheme, is avoided

The observables related to measurements by GRACE and discharge from gauging stations are the total mass anomaly $dM^T$ and the river runoff $R^R$. GRACE observations with acceptable error are still limited to monthly resolution. Discharge as well as some of the meteorological inputs like precipitation, evapotranspiration or moisture flux divergence are often measured in daily values, some of the products in monthly values. For an optimal adaption to the monthly resolution of GRACE products,
the approach presented here is based on monthly values but could also be applied to daily data without problems. The mass values used in the calculations here are assigned to the interval boundaries while the values for monthly recharge and measured runoff are constant over the interval and temporally assigned to the centre of the interval. Thus, for a comparison of the calculated mass and runoff values versus the observed monthly values of GRACE and discharge the calculated values have to be averaged over the interval. As the dynamics follow an exponential behaviour the mean values
cannot be taken from arithmetic averages at the interval boundaries but instead from an integral average over the interval.

The mean storage mass for $M_X$ is given for each interval $\lfloor t_i, t_{i+1} \rfloor$ by :

$$\overline{M}_{i+1}^{X} = \frac{1}{t_{i+1} - t_i} \int_{t_i}^{t_{i+1}} M_{i+1}^{X}(t - t_i) \cdot dt \tag{14}$$

leading to mean runoff $\quad \overline{R}^{X}(t) = \frac{1}{\tau_X} \cdot \overline{M}^{X}(t) \tag{15}$

i.e. mean catchment mass and runoff :

$$\overline{M}_{i+1}^{C} = \left(M_i^{C} - N_{i+1} \cdot \tau_C\right) \cdot \frac{\tau_C}{(t_{i+1} - t_i)} \left(1 - e^{-\frac{t_{i+1} - t_i}{\tau_C}}\right) + N_{i+1} \cdot \tau_C \tag{16}$$

and $\quad \overline{R}_i^{C} = \frac{1}{\tau_C} \cdot \overline{M}_i^{C} \tag{17}$

and mean river mass and runoff :

$$\overline{M}_{i+1}^{R} = \left(M_i^{R} - N_{i+1} \cdot \tau_R\right) \cdot \frac{\tau_R}{\left(t_{i+1} - t_i\right)} \cdot \left(1 - e^{-\frac{t_{i+1} - t_i}{\tau_R}}\right) + N_{i+1} \cdot \tau_R$$

$$+ \frac{\left[M_i^{C} - N_{i+1} \cdot \tau_C\right]}{\left(t_{i+1} - t_i\right)} \cdot \frac{\tau_R}{\tau_C - \tau_R} \cdot \left(\tau_R \cdot e^{-\frac{t_{i+1} - t_i}{\tau_R}} - \tau_C \cdot e^{-\frac{t_{i+1} - t_i}{\tau_C}} + \left(\tau_C - \tau_R\right)\right) \tag{18}$$

The Observables, which allow a comparison to measured data are :

- average river runoff $\overline{R}_i^{R} = \frac{1}{\tau_R} \cdot \overline{M}_i^{R}$ corresponding to measured monthly runoff $\tag{19}$

- average total mass $\overline{M}_i^{T} = \overline{M}_i^{C} + \overline{M}_i^{R}$ corresponding to monthly GRACE data $\tag{20}$

The equations Eq. (10) - Eq. (20) are self-consistent, i.e. the corresponding balance equations are fulfilled with :

$$\frac{M_{i+1}^{T}(t - t_i) - M_i^{T}}{(t - t_i)} + \overline{R}_{i+1}^{R}(t - t_i) = N_{i+1} \tag{21}$$

For the Single Storage approach the above piecewise analytical solutions of the Cascaded Storage approach, Eq. (8) - Eq. (21), are used for $\tau_R \ll \tau_C$ (here $\tau_R = 10^{-3}$ months). For this case the river network mass is negligible compared to the catchment mass

# 3 Properties and optimization performance

For the evaluation of the parameter optimization performance of the Cascaded Storage approach an example with synthetic recharge as input is investigated. This permits the quantification of the uniqueness and accuracy of the parameter estimation undisturbed by noise. It also facilitates the discrimination of errors in the calculation scheme itself and impacts arising from
undescribed processes when compared to real world data. For an application to GRACE measurements the main question is if and why the time constants $\tau_C$ and $\tau_R$ can be determined independently by an optimization versus anomalies in total mass and/or river runoff. Thus, in order to understand the optimization results with respect to uniqueness the general properties of the approach are presented and discussed first. For the synthetic case a recharge time series of sinusoidal form with a period of 12 arbitrary time units and length units with an amplitude and mean value of one is used as the driving force and the
calculation is run until equilibrium is reached. The example in Fig.2 shows the effect of a non negligible river network time constant $\tau_R =2.5$ time units for a catchment time constant $\tau_C = 3$ time units which leads to an increase in total mass $M^T(t) = M^C(t)+M^R(t)$ with respect to the average level and signal amplitude and to a phase shift between total mass $M^T$ and river mass $M^R$ i.e. the corresponding river runoff $R^R$.

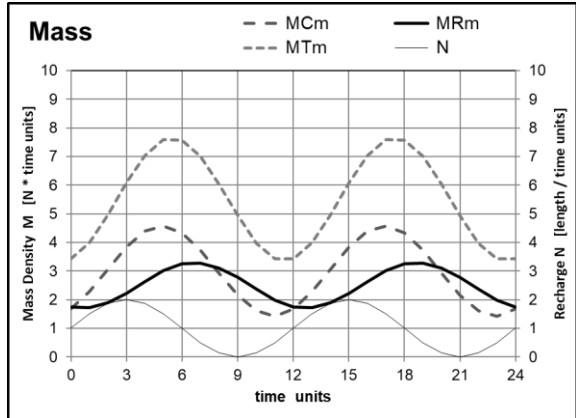

Fig.2: Time series of Recharge N, catchment mass $M^C$, river network mass $M^R$, and Total masses $M^T$ for the synthetic case at equilibrium

In order to describe the general behaviour of the mass and runoff time series and their dependence on $\tau_C$ and $\tau_R$, their properties are summarized here in the form of statistical values for the synthetic case with the sinusoidal recharge in equilibrium. This helps to understand why unique values for the time constants are achieved in the parameter optimization
process. The values of time constants $\tau_C$ and $\tau_R$ used for the statistical description cover a wide range from 0.1 to 100 time units and are combined independently.

## 3.1 Catchment and river mass

Based on the mean mass values, Eq.(14), (16), (18), of each stress period the long term averages for the storage compartments are given by :

$$\overline{M}^C = \overline{N} \cdot \tau_C \qquad \overline{M}^R = \overline{N} \cdot \tau_R \qquad \overline{M}^T = \overline{N} \cdot \left(\tau_C + \tau_R\right) \qquad\qquad (22a,b,c)$$

5 For $\tau_R \ll \tau_C$ (here $\tau_R = 10^{-3}$) the river network mass is negligible and the solution corresponds to a Single Storage approach. For a non negligible river network storage the given average values for total mass $M^T$ mean that the effective "total" time constant is given by the sum of the catchment and river time constants $\tau_T = \tau_C + \tau_R$, which means that the total mass $M^T$ observed by GRACE is bigger than the mass $M^{CC}$ calculated for the catchments alone. However, Equation (22c) cannot be used for the determination of $\tau_T = \tau_C + \tau_R$ from GRACE measurements directly as GRACE only provides mass anomalies.

The relative signal amplitudes (standard deviations normalized with those of the respective input) of both the catchment mass $M^C$ $^C$or river mass $M^R$ show the same functional form $\sigma_{MC} / \sigma_N \sim \sigma_{MR} / \sigma_{RC} = stdev(M^C) / N$ for the respective time constants $\tau_C$ or $\tau_R$ (Fig.3, $\tau_R = 10^{-3}$) with a monotonous increase to an asymptotic value $\sigma_{MC} / \sigma_N \sim \sigma_{MR} / \sigma_{RC} = 2$ which is reached at about one full period of the input. The superposition of the signal amplitudes for the observable total mass $M^T(t)$

15 $= M_C^C(t) + M^R(t)$ leads to a complex behaviour for $\sigma_{MT} / \sigma_N (\tau_C, \tau_R)$ (Fig.3), if the river time constant $\tau^R$ is not negligible ($\tau_R = 10^{-3}$) and especially if it gets close to $\tau_C$.

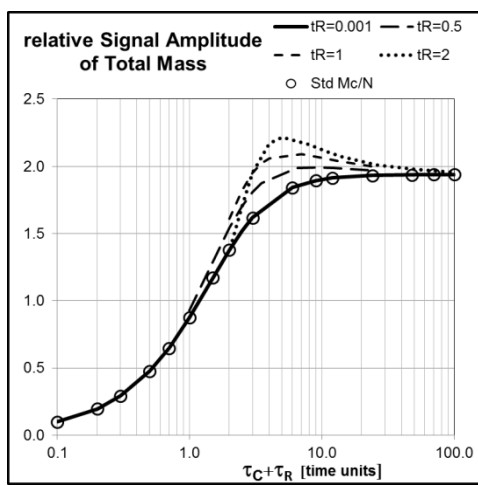

Fig.3: Signal Amplitudes of Total mass normalized by recharge: $\sigma_{MT} / \sigma_N$ versus total mass time constant $\tau_T = \tau_C + \tau_R$ for different river
20 time constants $\tau_R$

## 3.2 Catchment and river runoff

The calculated long term averages of the runoff contributions $R^C$ and $R^R$ correspond to the ones of the water balance equations, Eq.(4), (6), given by the mean recharge and thus are not dependent on the time constants.

$$\overline{R}^R(t) \;=\; \overline{R}^C(t) \;=\; \overline{N} \tag{23}$$

Thus, an observed long term average of runoff does not permit the determination of the time constant and hence the storage volume, Eq. (22).

The relative signal amplitudes of both, catchment and river runoff (normalized with the respective input $\sigma_{RC}/\sigma_N$ and $\sigma_{RR}/\sigma_{RC}$ show the same functional form corresponding to a Single Storage approach (Fig.4, $\tau_R = 10^{-3}$) and decrease monotonously with the respective time constants $\tau_C$ and $\tau_R$ to an asymptotic zero. However, the signal amplitude of the observable river

runoff $\sigma_{RR}/\sigma_N$ ($\tau_C + \tau_R$), normalized with recharge N, shows a deviations for different $\tau_C$ and $\tau_R$ with the same $\tau_C+\tau_R$ (Fig.4).

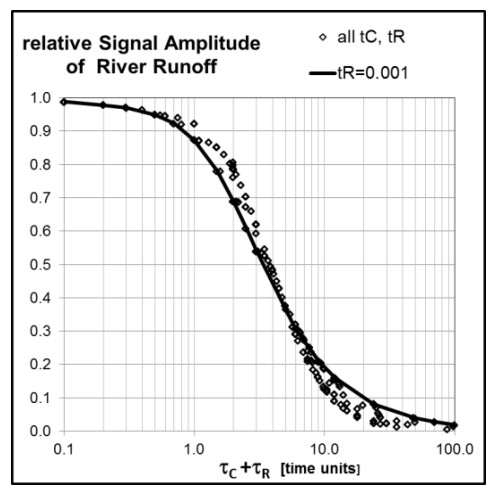

Fig.4: Signal Amplitudes for river runoff normalized by recharge: $\sigma_{RR}/\sigma_N$ versus total mass time constant $\tau_T = \tau_C + \tau_R$ for combinations in ($\tau_C$, $\tau_R$)

Both observables, total mass and river runoff, show a non unique behaviour with respect to combinations in ($\tau_C$, $\tau_R$) for the same $\tau_T = \tau_C + \tau_R$ and considerable deviations from the Single Storage approach ($\tau_R = 10^{-3}$). Measurements of the signal amplitudes thus only provide coarse estimates of the total time constant $\tau_T$, yet do not permit distinction between$\tau_R$ and $\tau_C$ and between catchment and river network storage.

However, so far, only the signal amplitudes are examined, but not the specific properties of the time series, i.e. the dynamic response to input signals in form and phase. The convolution in the solution of the balance equation, Eq.(8) and (11), leads to a different phasing with respect to the input N(t), which can be utilized for a separation of the respective time constants.

## 3.3 Phasing

For the synthetic example with a sinusoidal recharge time series N(t) as input the phasing ω of the different response signals is determined by the fit of a sinusoidal function (Fig.5). This facilitates the easy determination of the phasing and thus the relative phase shift Δω between the signals. Masses and the related runoffs are in phase for the same storage compartments, Eq.(15). For a negligible river network time constant ($\tau_R = 10^{-3}$) river runoff $R^R$ is in phase with the catchment storage $M^C$.

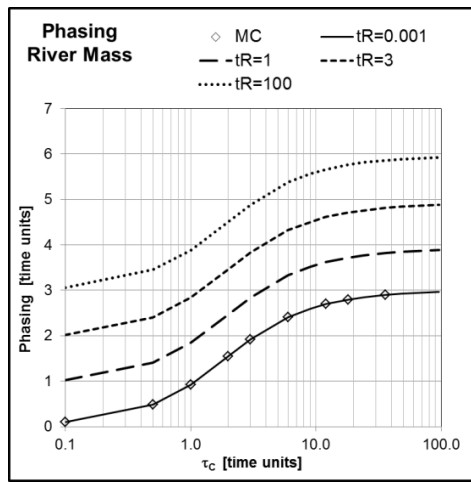

Fig.5: Phasing of river network mass with respect to recharge time series displayed versus $\tau_C$ for different $\tau_R$

The functional form of the phasing $\omega_M^C$ for the catchment mass $M^C$ $^C$or the corresponding runoff $R^C$ relative to recharge N(t) (Fig.5) can be empirically described by the monotonous function :

$$\omega_{MC}(\tau_C) = \omega_{max}\left(1 - e^{-\frac{\tau_C}{\lambda}}\right) \tag{24}$$

with the empirical parameters $\omega_{max} = 2,8$ and $\lambda = 2.7$ and an error $\varepsilon < \sim 2\%$ relative to the maximum.

As the catchment runoff $R^C$ with the phasing $\omega_M^C$ serves as input into the river system, the phasing of the river system with respect to to catchment runoff $R^C$, which has the same functional form as Eq. (24), is added on top of it (Fig.5). The resulting phasing of the river network storage or river runoff is thus given by a superposition in the form:

$$\omega_{RR}(\tau_C, \tau_R) = \omega_{max}\left(1 - e^{-\frac{\tau_C}{\lambda}}\right) + \omega_{max}\left(1 - e^{-\frac{\tau_R}{\lambda}}\right) \tag{25}$$

for any combination ($\tau_C$, $\tau_R$) and with the same empirical parameters as in Eq. (24).

As total mass $M^T(t) = M^{C C}(t) + M^R(t)$ is the superposition of the signals with the respective amplitudes and phasing, the phasing of total mass $M^T(t)$ is situated between catchment and the river system mass according to $\tau_R$. This means that for non negligible river network mass ($\tau_R > 0$) a phase shift between total mass (GRACE) and observed river discharge and also between total mass and modelled catchment mass must occur. The phasing of total mass $M^T(t)$ for all combinations ($\tau_C$, $\tau_R$) Fig.6 shows the same functional form as $\omega_{MC}$ and $\omega_{MR}$, Eq.(24), (25) if displayed versus the total time constant $\tau_T = \tau_C + \tau_R$.

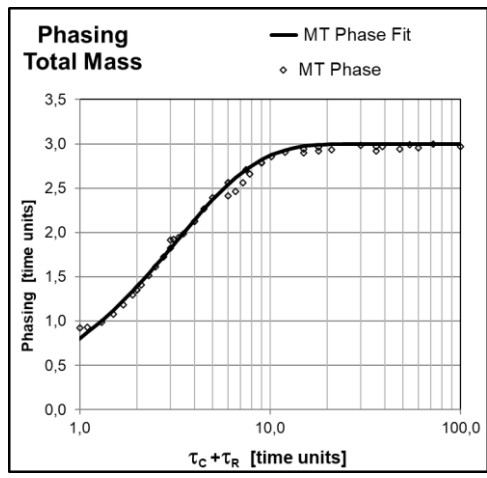

Fig.6: Phasing of Total mass versus total time constant $\tau_T = \tau_C + \tau_R$

It can be approximated by the fitting function $M^T$ fit :

$$\omega_{MT}(\tau_C, \tau_R) = \omega_{\max}\left(1 - e^{-\frac{\tau_C + \tau_R}{\lambda}}\right) \tag{26}$$

with the empirical parameters $\omega_{\max} = 2{,}95$ and $\lambda_T = 3.2$

The phase shift between GRACE total mass and river runoff is thus given by :

$$\Delta\omega(\tau_C, \tau_R) = \omega_{RR} - \omega_{MT} = \omega_{\max}\left(1 - e^{-\frac{\tau_C}{\lambda}}\right) + \omega_{\max}\left(1 - e^{-\frac{\tau_R}{\lambda}}\right) - \omega_{\max}\left(1 - e^{-\frac{\tau_C + \tau_R}{\lambda}}\right) \tag{27}$$

The empirical phase shift $\Delta\omega$ from Eq.27 corresponds to the one determined by a phase adaption $\Delta\omega_{adapt}$ (Eq. 38 & 39) of total mass and runoff within $<\sim 5\%$ (see supplement). This in principle allows for a determination of $\tau_C$ and $\tau_R$ separately from the adapted phase shift $\Delta\omega_{adapt}$ and the total mass time constant $\tau_T = \tau_C + \tau_R$ according to Eq. (27). However, errors

introduced by the linear interpolation used for the adaption of the phase shift lead to a much lower accuracy than the parameter estimation via the time series.

## 3.4 Parameter estimation

The analytical solutions for synthetic recharge time series permit the evaluation of the uniqueness and accuracy of the parameter optimization for given observables independent from limitations in the accuracy of numerical schemes and independent from noise in real world data sets. For given combinations ($\tau_C$, $\tau_R$) the analytical solutions are used as synthetic measurements and are fitted with the same algorithm in order to retrieve the fit parameters ($\underline{\tau}_C$, $\underline{\tau}_R$).

As the total mass $M^T$, Eq.(20), and the phasing, Eq.(25-27), are commutative in ($\tau_C$, $\tau_R$), either the data range $\tau_R < \tau_C$ or $\tau_R >$
$\tau_C$ has to be used for a unique optimization. This is realized via an additional constraint in the optimization. For the discussion here the condition $\tau_R < \tau_C$ is used, which hydrologically reflects the more frequent situations that the inundation volume is smaller than the catchment storage but the results can also be applied to $\tau_R > \tau_C$, which might be the case in flat areas with a dense river network (such as the Amazon), which typically leads to temporarily inundated areas.

As absolute signal values are not relevant for the determination of the time constant from runoff or not available for GRACE
data, the optimization versus the respective time series is based on signal amplitudes and the phasing. Thus, for a unique determination of ($\underline{\tau}_C$, $\underline{\tau}_R$) the following conditions have to be fulfilled:

a)  Optimization versus runoff

$$\sigma_{RR} / \sigma_N (\widehat{\tau}_C, \widehat{\tau}_R) = \sigma_{RR} / \sigma_N (\tau_C + \tau_R) \tag{28}$$

$$\omega_{RR}(\widehat{\tau}_C, \widehat{\tau}_R) = \omega_{max}\left(1 - e^{-\frac{\tau_C}{\lambda}}\right) + \omega_{max}\left(1 - e^{-\frac{\tau_R}{\lambda}}\right) \tag{29}$$

b)  Optimization versus mass anomalies

$$\sigma_{MT} / \sigma_N (\widehat{\tau}_C, \widehat{\tau}_R) = \sigma_{MT} / \sigma_N (\tau_C, \tau_R) \tag{30}$$

$$\widehat{\omega}_{MT}(\widehat{\tau}_C, \widehat{\tau}_R) = \omega_{MT}(\tau_C + \tau_R) = \omega_{max}\left(1 - e^{-\frac{\tau_C + \tau_R}{\lambda}}\right) \tag{31}$$

With the constraints $\tau_R < \tau_C$ or $\tau_R > \tau_C$ there is only one ($\underline{\tau}_C$, $\underline{\tau}_R$) fulfilling the respective conditions, thus leading to unique solutions. The optimization delivers RMSE errors for the time series in the range $10^{-8}$ - $10^{-7}$ and estimated time constants ($\underline{\tau}_C$, $\underline{\tau}_R$) with a relative error $\varepsilon(\underline{\tau}_X)/\tau_X$ which does not depend on absolute values of ($\tau_C$, $\tau_R$) but on their ratio $\tau_R/\tau_C$ (Fig.7).

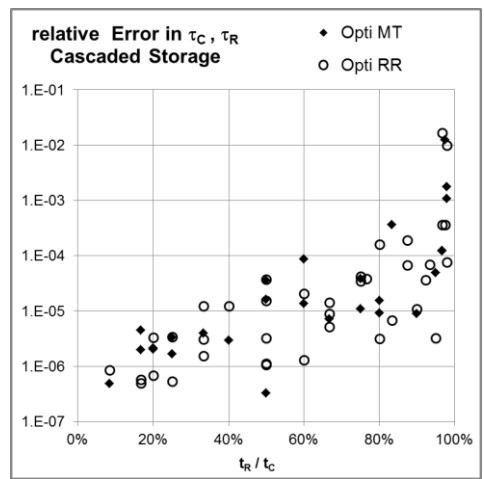

Fig.7: Relative error $\varepsilon(\underline{\tau}_X)/\tau_X$ of the time constants $\tau_C$ and $\tau_R$ for the Cascaded Storage approach with respect to optimizations versus Total mass $M^T$ or versus river runoff $R^R$.

For the synthetic case relative errors $\varepsilon(\tau_X)/\tau_X$ are very small (~$10^{-7}$ at $\tau_R/\tau_C \sim 0$) and show an exponential increase to a maximum of ~ 1% at $\tau_R \sim \tau_C$. The error for $\tau_R < \tau_C$ is analogous to $\tau_R > \tau_C$ and equal for an optimization versus runoff or mass anomalies .

For catchments showing a phase shift between total mass and runoff the description of the system by a Single Storage approach ($\tau_R = 10^{-3}$) leads to a considerably higher relative error $\varepsilon(\underline{\tau}_X)/\tau_X$ in the estimated time constant $\underline{\tau}_C \sim (\tau_C + \tau_R)$ and thus also in Drainable Storage volume. It follows a power function and corresponds to $\varepsilon < 10\%$ for $\underline{\tau}_C < 3$ and $\varepsilon > 40\%$ for $\underline{\tau}_C > 6$. For this case the optimization versus river runoff or mass anomalies leads to different total time constants (rel. Diff. $\varepsilon > 7\%$ for $\underline{\tau}_C > 5$). Even though this might look like an acceptable result for $\underline{\tau}_C < 3$, there are still inevitable deviations in signal amplitudes (10-20%) and phasing between the modelled and measured signals for both total mass and river runoff time series.

It can be summarized that in contrast to the Single Storage approach the Cascaded Storage approach permits the determination of both time constants ($\tau_C$, $\tau_R$) independently in a unique, highly accurate way for optimizations with respect to either total mass anomalies or river runoff. However, it has to be mentioned that even though the theoretical error in time constants remains below 1% for $\tau_R \sim \tau_C$, the ambiguity for $\tau_R < \tau_C$ or $\tau_R > \tau_C$ cannot be solved without further information on the volume of the river network.

## 3.5 Fully data driven Determination of Drainable Storage volumes

For the case that river discharge is available for a sufficient period of time the Cascaded storage approach facilitates a simple determination of the Drainable water storage volumes both for the catchment and for the river network directly from observations without the necessity of new model runs. The two time constants ($\tau_C$, $\tau_R$) adapted during a training period permit to quantify the Drainable water storage volumes $M^T$, $M^{CC}$ and $M^R$ at other times directly from observations of GRACE mass anomalies and river discharge. With a simple numerical adaption of the phase shift $\Delta\omega$ resulting from the time constants ($\tau_C$, $\tau_R$) according to EQ.27 a quite accurate determination of the total Drainable storage volume from measured river discharge exclusively or the other way round of runoff from GRACE mass anomalies is possible.

These can be determined by the following calculations :

1. Long term averages of Drainable storage volumes from observed runoff $R_o$ :

$$\overline{M}_{sim}^{C} = \tau_C \cdot \overline{R}_o \qquad \overline{M}_{sim}^{R} = \tau_R \cdot \overline{R}_o \qquad \overline{M}_{sim}^{T} = \tau_T \cdot \overline{R}_o \qquad \text{with} \quad \overline{R}_o = \overline{N} \quad \text{and } \tau^T = \tau^C + \tau^R$$

according to Eq. (22a,b,c) and Eq. (23) :

2. Time series of Drainable storage volumes from GRACE and observed runoff $R_o$
   without the need for a phase adaption:

$$M_{sim}^{R}(t) = \tau_R \cdot R_o(t) \tag{35}$$

($NS_S$ 0.961, $NS_R$ 0.576, corr$_R$ 0.859  vs $M^R$ from Eq.18)

$$M_{sim}^{T}(t) = dM^T(t) + \overline{M}^T = GRACE(t) + \overline{M}^T = GRACE(t) + \tau_T \cdot \overline{N} \cdot = GRACE(t) + \tau_T \cdot \overline{R}_0 \tag{36}$$

($NS_S$ 0.973, $NS_R$ 0.751, corr$_R$ 0.901  vs $M^T$ from Eq.20)

$$M_{sim}^{C}(t) = M^T(t) - M^R(t) = GRACE(t) + \tau_T \cdot \overline{R}_o \; - \; \tau_R \cdot R_o(t) \tag{37}$$

($NS_S$ 0.906, $NS_R$ -0.065, corr$_R$ 0.607  vs $M^{CC}$ from Eq.16)

The simplified calculations directly based on observations lead to accurate equivalences to the fully calculated time series of total and the river network storage volumes $M^T$ and $M^R$ and to a reasonable description of the catchment storage volumes $M^C$.

3. Time series of total Drainable storage volumes $M^T$, directly from observed runoff $R_o$

or simulated river runoff $R^R_{sim}$ from GRACE with a numerical phase adaption of $\Delta\omega$.

Use of the phase shift $\Delta\omega_{adapt}$ adapted between GRACE and observed river runoff by a linear temporal interpolation Riegger and Tourian (2014) permits a simple description of river runoff directly from GRACE (Eq.38) or of total Drainable water storage $M^T$ directly from observed runoff (Eq.39) and corresponds to $\Delta\omega$ from Eq.27 within $<\sim 10\%$. Both lead to very similar fitting performances.

$$R^R_{sim}(t_i) \; = \; \frac{1}{\tau_T} \cdot \left[ \; (1-\Delta\omega)\cdot GRACE(t_i) \; + \; \Delta\omega \cdot GRACE(t_{i-1}) \; + \; \tau_T \cdot \overline{R}_o \right] \qquad (38)$$

(NS$_S$ 0.943, NS$_R$ 0.698, corr$_R$ 0.864 vs measured R$_o$,)

$$M^T_{sim}(t_i) = \tau_T \cdot \left[ \; (1-\Delta\omega)\cdot R_o(t_i) \; + \; \Delta\omega \cdot R_o(t_{i+1}) \; \right] \qquad (39)$$

(NS$_S$ 0.946, NS$_R$ 0.483, corr$_R$ 0.859 vs GRACE)

For the representativeness of the fitting performance the fully data driven approach (Eq.35-Eq.39) is compared to the respective masses and runoff from the Cascaded storage approach applied to the Amazon basin (see below) and not to synthetic data. The related calculations are accessible in the xls-workbook provided in the supplement.

This performance means that the determination of the two time constants ($\tau_C$, $\tau_R$) by the Cascaded storage approach during a sufficient training period facilitates a simple quantification of Drainable storage volume or runoff time series directly from measured river discharge or GRACE anomalies. This provides a possibility to close data gaps in river discharge or GRACE directly from measurements with high accuracy.

# 4 Application to the Amazon basin

The R-S diagram of the full Amazon basin shows a hysteresis (Fig.1b, d) corresponding to a phase shift, which can be interpreted as the time lag of river discharge. The Amazon basin upstream Obidos is situated in a fully humid tropic environment with permanent, yet variable recharge and is large enough ($4704394 km^2$) for low noise levels in the signals of GRACE and moisture flux divergence. With permanent recharge flow contributions from overland flow and groundwater cannot be distinguished in the discharge curve. Also, on a spatial average over the full Amazon basin, with permanent recharge the uncoupled storages (like soil water storage, open water bodies etc.) are not time variant, i.e. there is no dry out effect. Any contribution from time dependent, uncoupled storages could be recognized in the R-S diagram as it would appear

as a hysteresis, which does not correspond to a time lag, or by the respective deviations in the scatter plots of calculated versus measured runoff or storage volumes (see supplement) . This is not the case.

Generally recharge from different approaches and products can serve as input to the system such as :

1.        $N(t) \; = \; P(t) - ET(t)$                                                            (32)

         from the hydrometeorological products precipitation P and actual evapotranspiration ETa

   2.        $N(t) \; = -\nabla \cdot \vec{Q}$                                                                  (33)

         from atmospheric data, with monthly vertically integrated moisture flux divergence viMFD

   3.        $N(t) = \dfrac{\partial}{\partial t} M(t) + R(t)$                                              (34)

10          from the terrestrial water balance with monthly temporal derivatives of GRACE measurements and measured river runoff $R_o$ of the basin.

Here recharge [mm/month] is taken either from the water balance, Eq.(34), or from moisture flux divergence, Eq.(33), provided by ERA-INTERIM of ECMWF and processed by the Institute of Meteorology and Climate Research, Garmisch,
Germany. For GRACE mass anomalies data from GeoForschungsZentrum GFZ Potsdam Release 5 are used in mm equivalent water height. Both are handled as described in detail in Riegger and Tourian (2014). Their spatial resolution limits the application of the approach to global scales >>200000km$^2$. River discharge is taken from the ORE HYBAM project (http://ore-hybam.org) and converted to runoff [mm/month] by normalization with the basin area. For a comparison of the calculated river network storage with observations from the "Global Inundation Extent from Multi-Satellite GIEMS Prigent
et al. (2001) flood area [km$^2$] is used. As GRACE mass anomalies are most accurate for a monthly time resolution at present, the other data sets are aggregated to a monthly resolution as well. For the parameter optimization time series of river runoff and GRACE mass anomalies are used for the time period from January 2004 until January 2009. Monthly runoff and the storage volume of the basin and river network are calculated for Amazon based on different recharge products here and optimized either versus runoff or GRACE mass anomalies . The results calculated with recharge from the terrestrial water
balance optimized versus GRACE are shown in Figures 8-10 for both (a) the monthly signal and (b) the monthly residual (monthly value minus mean monthly value) for January 2003-2009 .

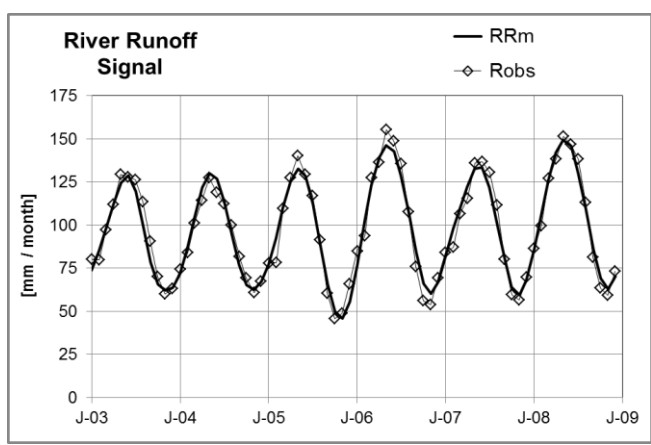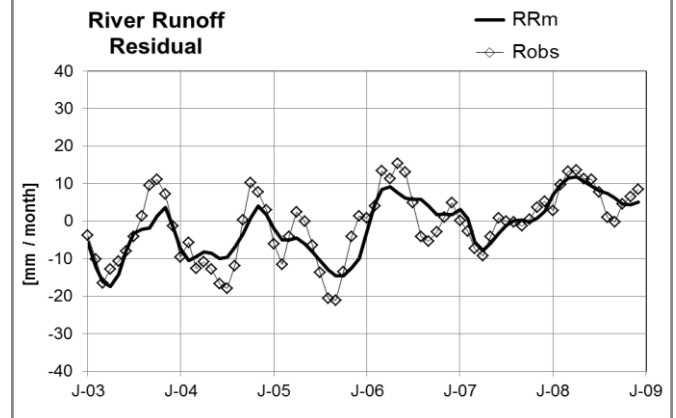

Fig.8: Time series of river runoff $R^R_m$ for the Amazon basin (Obidos) and optimization versus GRACE (a) for the signal (b) for the residual

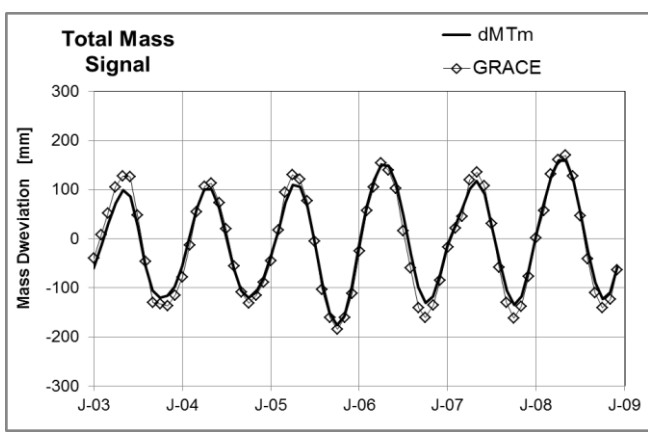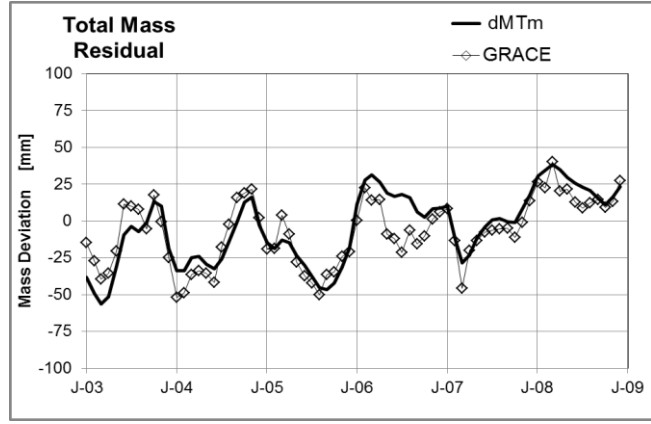

5  Fig.9: Time series of Total mass anomalies $dM^T$ for the Amazon and optimization versus GRACE (a) for the signal (b) for the residual

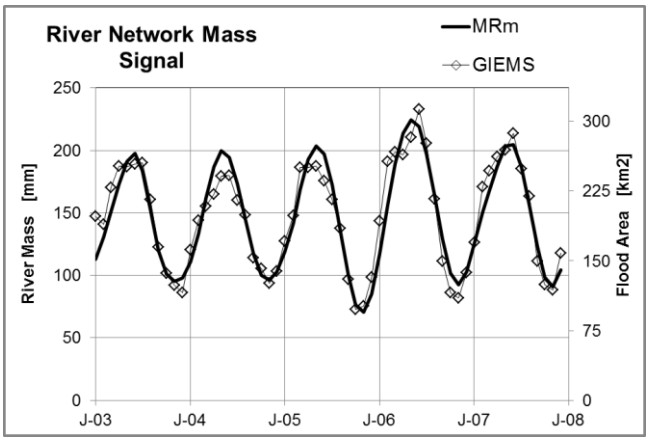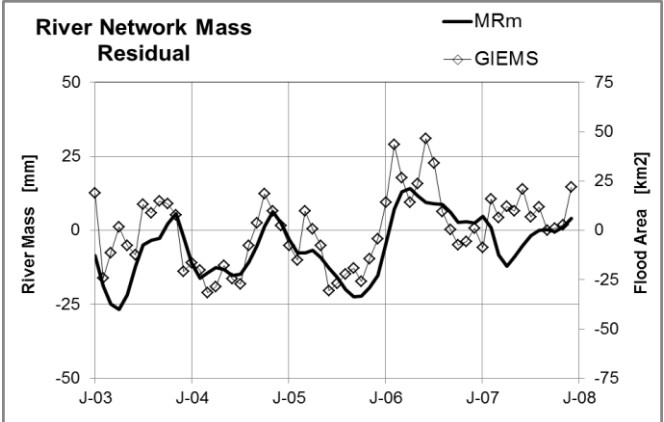

Fig.10: Time series of River network storage $M^R_m$ and inundated area from GIEMS for the Amazon (a) for the signal (b) for the residual

The calculated river runoff $R^R$, total mass anomaly $dM^T$ and river network mass $M^R$ fit very well with the measured river runoff, GRACE and the flooded area from GIEMS both with respect to the signal and the de-seasonalized monthly residual. The Cascaded Storage approach reproduces the phase shift between measured runoff $R_o$ and total mass $dM^T$ (or GRACE respectively). The calculated river network mass $M^R$ of about 50% of the total mass $M^T$ for Amazon is proportional to observed runoff $R_o$ without any phase shift (Fig.11)!

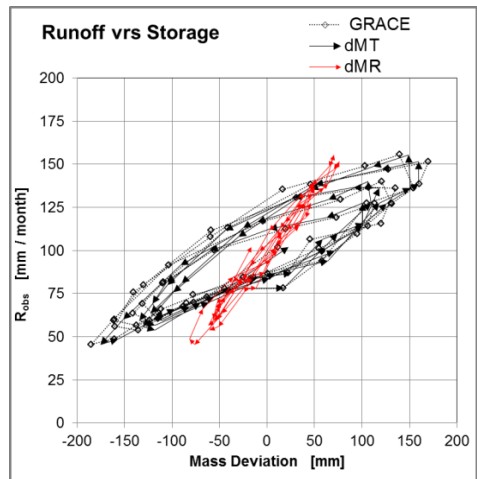

Fig.11: R-S relationships for observed runoff $R_o$ versus the mass anomalies of GRACE, calculated Total mass $dM^T$ and river network mass $dM^R$

Calculated hydraulic time constants, mean values and signal amplitudes for the absolute storage volumes are provided in Table.2 for the full Amazon basin upstream Obidos. In addition the performance of optimizations either versus river runoff (Column A) or versus GRACE (Column B) and for different recharge products (Column D, E) is displayed. This shows that the optimization versus different references leads to a very similar results while the fitting performance for the two recharge products (Columns A, B and D, E) is quite different. For recharge from water balance, Eq.(34), the resulting time constants and thus the storage masses differ in a range of ~5% for the different references while they vary ~10% for recharge from moisture flux divergence.

In order to illustrate the benefits of the Cascaded versus a Single Storage approach even in the fitting quality, results for a fixed $\tau_R = 10^{-3}$, which correspond to a Single Storage, are shown (Column C, F) for different recharge products. With the Single Storage approach - beside the much worse fitting performance - the resulting time constant $\tau_T = \tau_C + \tau_R$ is overestimated (corresponding to the investigations in section 3) and the modelled signal amplitude is about 20% less than that measured from GRACE. In addition a non negligible phase shift remains between the modelled runoff and measured discharge.

| | **A** | **B** | **C** | **D** | **E** | **F** |
|---|---|---|---|---|---|---|
| **Approach** | Cascaded | Cascaded | Single | Cascaded | Cascaded | Single |
| **Recharge** | R+dM/dt | R+dM/dt | R+dM/dt | -divQ | -divQ | -divQ |
| **Optimization** | RR | dMT | dMT | RR | dMT | dMT |
| $\tau_C$ [month] | 1.53 | 1.62 | 3.55 | 1.68 | 1.87 | 3.95 |
| $\tau_R$ [month] | 1.53 | 1.62 | 0.001 | 1.68 | 1.87 | 0.001 |
| **Avg $M^T$ [mm]** | 304.81 | 321.77 | 353.29 | 333.00 | 370.93 | 392.02 |
| **Avg $M^{CC}$ [mm]** | 152.17 | 160.58 | 353.19 | 166.23 | 185.12 | 391.92 |
| **Avg $M^R$ [mm]** | 152.64 | 161.18 | 0.10 | 166.77 | 185.81 | 0.10 |
| **Avg $R^R$ [mm month$^{-1}$]** | 99.53 | 99.59 | 99.39 | 99.35 | 99.49 | 99.34 |
| **Avg $N$ [mm month$^{-1}$]** | 98.80 | 98.80 | 98.80 | 99.07 | 99.07 | 99.07 |
| **Stdev $M^T$ [mm]** | 98.46 | 100.38 | 84.09 | 101.73 | 105.34 | 87.83 |
| **Stdev $M^C$ [mm]** | 58.49 | 60.40 | 84.06 | 61.22 | 65.05 | 87.80 |
| **Stdev $M^R$ [mm]** | 45.48 | 46.02 | 0.02 | 46.70 | 47.55 | 0.02 |
| | | | | | | |
| **RMSE $R^R$-$R_o$ [mm month$^{-1}$]** | 5.76 | 6.08 | 12.13 | 11.99 | 12.57 | 18.08 |
| **RMSE $M^T$-GRACE [mm]** | 15.28 | 14.73 | 28.93 | 35.45 | 34.54 | 42.31 |
| **$NS_S$ $R^R$-$R_o$** | 0.96 | 0.96 | 0.84 | 0.85 | 0.83 | 0.65 |
| **$NS_R$ $R^R$-$R_o$** | 0.74 | 0.72 | 0.73 | -0.09 | -0.08 | -0.10 |
| **$corr_S$ $R^R$-$R_o$** | 0.98 | 0.98 | 0.94 | 0.92 | 0.92 | 0.82 |
| **$corr_R$ $R^R$-$R_o$** | 0.86 | 0.85 | 0.87 | 0.48 | 0.46 | 0.41 |
| **$NS_S$ $dM^T$-GRACE** | 0.98 | 0.98 | 0.92 | 0.89 | 0.89 | 0.84 |
| **$NS_R$ $dM^T$-GRACE** | 0.74 | 0.72 | 0.71 | -0.57 | -0.81 | -0.71 |
| **$corr_S$ $dM^T$-GRACE** | 0.99 | 0.99 | 0.98 | 0.94 | 0.94 | 0.93 |
| **$corr_R$ $dM^T$-GRACE** | 0.90 | 0.90 | 0.88 | 0.58 | 0.56 | 0.51 |
| **$corr_S$ dGIEMS-GRACE** | 0.92 | 0.92 | 0.92 | 0.92 | 0.92 | 0.92 |
| **$corr_S$ GIEMS-$M^T$** | 0.93 | 0.94 | 0.95 | 0.82 | 0.84 | 0.82 |
| **$corr_S$ GIEMS-$M^R$** | 0.96 | 0.95 | 0.95 | 0.88 | 0.86 | 0.82 |
| **$corr_R$ dGIEMS-GRACE** | 0.65 | 0.65 | 0.65 | 0.65 | 0.65 | 0.65 |
| **$corr_R$ GIEMS-$M^R$** | 0.76 | 0.75 | 0.78 | 0.04 | -0.01 | 0.01 |

Table.2: The statistical characteristics are listed for calculated river runoff $R^R$, total mass $M^T$, basin mass $M^C$ and river network mass $M^R$ and observed river runoff $R_o$, GRACE mass anomalies and flood areas from GIEMS using: RMSE: Root-mean-square error of simulated versus measured, $NS_S$: Nash Sutcliffe coefficient of the signal (simulated values versus long-term mean of measured), $NS_R$: Nash Sutcliffe coefficient of monthly residuals (simulated values versus monthly mean of measured), $corr_S$: correlation of simulated versus measured signals, $corr_R$: correlation of simulated versus measured monthly residuals, Avg and Stdev are the long term mean and standard deviations, prefix "d" used for anomalies related to the long term mean. Results are compared for the different optimization references runoff $R_o$ or GRACE for recharge from water balance R+dM/dt (A, B) and for atmospheric input -divQ (D, E). The Cascaded storage approach is compared to Single storage approach in (C, F).

The Cascaded Storage approach with recharge from the water balance, Eq. (34), leads to high accuracy fits between calculated and measured river runoff and total storage mass for the signals ($NS_S$ $R^R$-$R_o$=0.96, $NS_S$ $dM^T$- GRACE = 0.98) and for the residuals ($NS_R$ $R^R$-$R_o$ = 0.74, $NS_R$ $dM^T$- GRACE = 0.74, the respective calculations are available in the xls-workbook provided in the supplement).

The comparison of the water budgets for 14 different GHMs / LSMs Getirana et al. (2014) for the Amazon basin permits to sort in the results of the Cascaded Storage Approach into those of the GHMs/LSMs Getirana et al. (2014), (Fig. 14). With a Nash-Sutcliffe coefficient $NS_R$ (R = with respect to the mean seasonal cycle) of 0.74 and a correlation $corr_R$ = 0.90 compared to an $NS_R$ of 0.58 and a correlation $corr_R$=0.84 for the best LSM, the Cascaded Storage approach outperforms the GHMs/LSMs for full Amazon. Even the fully data driven approach (Eq.39) with an $NS_R$ of 0.483, and a correlation
$corr_R$=0.859 for simulated mass anomaly versus GRACE is comparable to the best performances in the GHM/LSM test by Getirana et al. (2014). The related calculations are accessible in the xls-workbook provided in the supplement.

This is partly seen as the result of the simplicity of the lumped approach averaging out errors that emanate from the large number of different processes described by the GHMs/LSMs. However, the main reason for the better performance is seen in the quality of recharge data taken from the water balance using GRACE and river runoff, as the use of moisture flux
divergence for this purpose leads to much worse performance.

The calculated river network mass $M^R$ of the Amazon varies in the range of 40-65% of total mass $M^T$ with an average ~50%, corresponding to the values found by Paiva et al. (2013) and Papa et al. (2013) or ~41% by Getirana et al. (2017a). The correlation versus the observed flood area from GIEMS is higher for the calculated river network mass $M^R$ (0.96 for the
signal and 0.76 for the monthly residual) than for GRACE (0.92 and 0.65 respectively). The consistency of the calculated river network mass (and the corresponding observed river runoff $R^R = M^R \tau_R^{-1} = 0.742\ M^R$) with the flood areas is seen much more clearly in the phasing (Fig.12), which shows a clear phase shift for GRACE versus GIEMS (see also Papa et al. (2008)), yet none for calculated $M^R$. As already Getirana et al. (2017a) emphasized, only an appropriate description of the river network storage permits a correct description of the total storage in amplitude and phasing for a comparison with
GRACE.

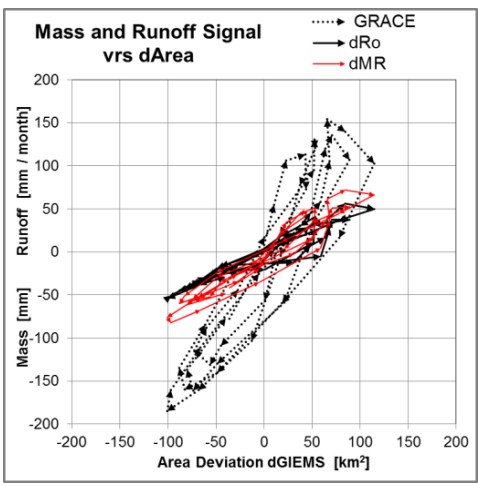

Fig.12: GRACE mass, calculated river network mass $dM^R$ and observed river runoff $dR_o$ versus Flood Area dGIEMS; all displayed as anomalies (please consider $dM^R = dR^R\tau^R = 1.53\ dR^R$ )

# 5 Discussion

Distributed hydrological models use a lot of detailed local information in order to address a large number of involved processes for each grid cell. In this way they provide a spatially distributed and a very detailed composition of the involved storages and flows. However, it is very difficult to discriminate the respective processes locally with the consequence that only their superposition can be compared to measurable data like river discharge. This creates a kind of ambiguity between the different contributions, thus loosing parts of the benefits of a detailed description. As it has also been pointed out by

Getirana et al., 2017a, for a comparison of the superposed storages to GRACE anomalies the river network storage changes have to be quantified as well, as only the total storage changes are measured by GRACE. This means that an appropriate description of the river network storage and the time lag is an inevitable prerequisite for an appropriate adaption of model parameters. A hydrodynamic modelling of the river network facilitates the quantification of its storage to be sure, yet means a real computational challenge.

The Cascaded storage approach permits the quantification of the Drainable storage volumes both for land masses and river network directly from GRACE and river discharge for gauged catchments or from GRACE and recharge for ungauged catchments by adapting only two parameters, the time constants. No detailed information on local vegetation, surface, unsaturated / saturated zone, etc. and related flow processes nor a hydrodynamic modelling with detailed hydraulic

information on river roughness, cross section, gradient or backwater effects is needed.

At present, the Cascaded storage approach is limited to climatic and physiographic conditions for which the hysteresis is completely explained by a time lag. i.e. that no impacts of uncoupled storage components are visible in the R-S diagram. For

a global coverage the Cascaded storage approach has to be extended by an explicit integration of coupled and uncoupled storage compartments to account for other regional climatic and physiographic conditions. The uncoupled storage components then have to be quantified either by their absolute storage volume or by their relative contribution to total storage.

As Riegger and Tourian, 2014, have shown for boreal catchments, this can be done by means of remote sensing and a conceptional description. Boreal catchments are temporarily dominated by snow leading to a huge hysteresis due to a superposition of masses from fully coupled (liquid) and uncoupled (solid) storage compartments. Remote sensing of the catchments snow coverage by MODIS facilitates the separation of the coupled liquid storage (proportional to river runoff) on the uncovered areas and the uncoupled frozen part on the snow covered areas. The coupled liquid storage determined in this way actually constitutes a LTI system, i.e. the hysteresis can be fully explained by a phase shift. This fulfils the prerequisites for the Cascaded storage approach and thus permits an application to boreal catchments as well. In consequence, the principle of the Cascaded storage approach is not limited to fully humid climatic conditions. It permits an application to other climatic regions as well, provided that the coupled and uncoupled storage compartments can be separated.

The description of monsoonal regions for example, which play an important role in the global water budget, is a considerable challenge. For these regions with seasonally dry periods high precipitation events during the wet season lead to distinct runoff in parallel from overland and groundwater with different time constants $\tau_{Surface}$ and $\tau_{GW}$ and to time dependent uncoupled storage compartments like soil or isolated open water bodies, which do not contribute to discharge. All these storages have to be addressed for an adequate description of the drainable storage volumes. The uncoupled storage compartments have to be quantified by remote sensing (soil moisture and open water body altimetry from satellites) and subtracted from the total catchment mass measured by GRACE, which of course is a major task.

# 6 Conclusions

The test of the Cascaded storage approach with synthetic recharge data has shown that the parameter optimization either versus mass anomalies or runoff reproduces the time constants ($\tau_C$, $\tau_R$) for both, the catchment and the river network in a unique way with high accuracy, yet with an ambiguity for $\tau_R < \tau_C$ or $\tau_R > \tau_C$, and thus in the related storage volumes. This problem can only be solved by reasonable assumptions or better by additional information on the volume of river network or flood areas, which can be taken from ground based observations or remote sensing like GIEMS flood areas and water levels from altimetry. Numerical tests have also shown that the description of a system (showing a phase shift) by a Single Storage approach can only address the total Drainable storage and thus leads to phasing differences between the calculated and measured runoff or storage and to considerable errors in the time constant of the total system.

The application to the full Amazon basin shows that the system behaviour including the time lag can be described by a simple conceptual model with a catchment and a river network storage in sequence and an adjustment of only two parameters, the time constants. The storage amplitudes for the total Drainable water storage and the time lag to runoff are described with high precision. Calculated river network volume and the observed flood area correspond to GIEMS observations and newest model results incl. river routing Getirana et al. (2017a) and are in phase with river discharge. This independent quantification of the river network volume permits an investigation of the relationship between flood areas, flood volumes, river runoff and calculated river network with its additional information and might provide insights into river hydraulics i.e. routing times and the mass- area- and level- relationships of flooded areas.

As the optimization performance is comparable for either reference, the observed river runoff or GRACE anomalies, a calculation with given recharge and an optimization versus measured GRACE data can be used to determine both, the river discharge as well as the Drainable storage volumes even for ungauged basins. For these cases the availability of accurate recharge data determines the accuracy of runoff and storage calculations at present. However, for ungauged basins the use of moisture flux divergence still provides quite acceptable results based on remote sensing and atmospheric data exclusively.

For the case that river discharge is available for a sufficient period of time in order to adapt the two time constants ($\tau_C$, $\tau_R$) sufficiently the Cascaded storage approach facilitates a simple "fully data driven" determination of the Drainable water storage volumes $M^T(t)$, $M^C(t)$ and $M^R(t)$ at other times directly from observations of GRACE or river discharge without the necessity of new model runs. This permits to close data gaps or even to make forecasts within the period of the time lag.

As the spatial resolution of GRACE and the accuracy of moisture flux divergence is limiting the applicability of the Cascaded storage approach to global scale catchments at the moment, any improvement in the spatial / temporal resolution and accuracy of GRACE and hydrometeorological data products will tremendously increase the number of catchments which can be described by this approach in future.

# Data availability

In the supplement calculations and data are provided in an EXCEL workbook for the synthetic case and for the Amazon catchment.

# Acknowledgements

The author would like to thank Nico Sneeuw and Mohammad Tourian from the Institute of Geodesy, Stuttgart, for the handling of GRACE data , Harald Kunstmann and Christof Lorenz from the Institute of Meteorology and Climate Research, Garmisch, for the provision of moisture flux data and Catherine Prigent, Observatoire de Paris, for the provision of Flood areas from the GIEMS project.

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
