# Peer review of ""Quantification of Drainable Water Storage Volumes on Landmasses and in River Networks based on GRACE and River Runoff using a Cascaded Storage Approach – First Application on Amazon""

_Hydrology and Earth System Sciences, 2018_

## Referee Comment (RC1) · M. Bierkens (Referee) · 11 May 2018

This is interesting work showing how GRACE and discharge data can be used to estimate drainable storage in river basins and river networks. Although the ideas are interesting and the data-analysis well developed, there are some major reservations I have with the approach and the paper.

1. I fail to see why knowing the drainable storage of catchments and river networks itself is so interesting. In applications of hydrology one is generally interested in discharge anomalies (high flow, low flow), evaporation anomalies (agricultural drought) and flooded areas. So, I feel that the necessity of this work should be re-stated. In fact,

the main, and very interesting, contribution is that GRACE data alone (together with recharge [precipitation surplus would be a better term] estimates from e.g. moisture convergence) can be used to estimated river discharge in ungauged basins.

2. Similarly, in stressing better the necessity of the approach, it should be made clear why large-scale hydrological models could not be used to do the job. Note that some of these models (such as WaterGap and PCR-GLOBWB) have groundwater parameterizations and are able to reproduce the amplitude and lags observed in GRACE (see e.g. Wada et al., 2012; Water Resour. Res., 48, W00L06). If the amplitude and phase shift between recharge and runoff are informative about storage and hence discharge, GRACE anomalies could be used to calibrate these models as well, with the added advantage that a) we do not need to assume linearity between storage and discharge; b) these models deal with temporarily unconnected storages as well.

3. Modelling the effect of the drainage networks as a linear storage-outflow relationship may be valid for the Amazon where during peaks the whole basin turns into a huge flooded area resembling a lake. But in many rivers of the world, e.g. the Danube, the Rhine, the Nile, water during high stages is confined in the channel or in narrow valleys and the lag between catchment discharge and discharge at the basin's outlet is more of a travel-time phenomenon then a storage attenuation phenomenon. In this case, a routing routine such as used in many global hydrological models would be more suitable, with the unknown parameters the channel and floodplain resistance parameters (e.g. Manning coefficient).

4. More generally: the approach seems to be valid in large humid basins, without cold-region processes, where all active stores are permanently connected to the discharge mechanism, while routing is such that the drainage network can be represented by a storage-outflow relationship. This makes the applicability of the approach somewhat limited.

5. The writing should definitely be improved. For instance, the abstract reads like an

extended summary with an introduction and is too long and too specific. Also, the use of the English language should be checked by a native speaker. Suggestions for improvements and some other small remarks are given in the annotated manuscript attached.

Please also note the supplement to this comment:
https://www.hydrol-earth-syst-sci-discuss.net/hess-2018-38/hess-2018-38-RC1-supplement.pdf

[Figure]

**Supplement:**

[revised manuscript text omitted]

---

## Author Comment (AC1) · 15 Jun 2018

< This is interesting work showing how GRACE and discharge data can be used to estimate drainable storage in river basins and river networks. Although the ideas are interesting and the data-analysis well developed, there are some major reservations I have with the approach and the paper.

-> I would like to thank the reviewer for his comments as they triggered some corrections and amendments in the manuscript. - Below you find my respond to his comments in detail.

< 1. I fail to see why knowing the drainable storage of catchments and river networks itself is so interesting. In applications of hydrology one is generally interested in discharge anomalies (high flow, low flow), evaporation anomalies (agricultural drought) and flooded areas. So, I feel that the necessity of this work should be re-stated.

-> The knowledge of Drainable water storages is essential for the description and prediction of river discharge (high flow, low flow), for water ecology, river management (floods and draughts) and especially for water resources management (seeP1 L8-10). This is what I consider as one of the main concerns in hydrology. The knowledge of storage volume changes in the form of seasonal amplitudes and trends, as provided by GRACE, does not help to assess available water resources and especially to quantify possible problems and conflicts with respect to future water supply (see Introduction P2 L13-P3 L21)). Time variable storage compartments, not contributing to river discharge like canopy or soil are certainly important for local ecology, but they are normally not relevant for water supply. Water extraction from groundwater storages in low permeable structures or from deep fossil resources means a lot of technical effort and energy input. Drainable storages comprise all water storages which are accessible with relatively little effort and energy input as for surface water or shallow groundwater systems. Their global scale quantification provides an effective overview on globally available water resources. The possibility to determine the drainable storage with the approach presented (see P 23 L26-33, Eq.22a,b,c, Eq 23, Eq35, EQ.36, Eq.38) allows to observe the storage status of global scale catchment directly by remote sensing from GRACE in a global distribution. This means a major step in environmental remote sensing.

< In fact, the main, and very interesting, contribution is that GRACE data alone (together with recharge [precipitation surplus would be a better term] estimates from e.g. moisture convergence) can be used to estimated river discharge in ungauged basins.

-> The possibility to optimize the approach versus GRACE only using moisture flux divergence as input is a very promising perspective for ungauged catchments. Yet as

the accuracy of the drainable water storage and the calculated river discharge depends on the quality of recharge data it is limited by the quality of moisture flux divergence at the moment (see P24 L3-11).

< 2. Similarly, in stressing better the necessity of the approach, it should be made clear why large-scale hydrological models could not be used to do the job.

-> The author does not claim that large scale models cannot be used to do the job. Global hydrological models are able to describe a large number of storages like canopy, soil zones, surface water, groundwater, river network etc. and describe their storage volumes and the related flows. This is a real benefit to understand details in the water cycle. However, one of the difficulties in verifying large scale hydrological models consists in the quantification of the individual storage volumes and related flows by ground based measurements. These are mainly point measurements with the necessity of an interpolation and with unknown storage coefficients (see P3 L8-16). GRACE anomalies now allow for a direct comparison of the measured total mass changes and the respective sum over all simulated storage compartments with respect to amplitudes and phasings. Comparisons of the simulated total storage versus GRACE show considerable differences for several models as for LAD (Milly and Shmakin, 2002), GLDAS (Rodell et al., 2004) and WGHM (Döll, Kasper and Lehner, 2003), just to mention a few. For the different models an underestimation of the signal amplitudes and phase shifts between measured and simulated total mass is reported (Güntner et al., 2007, Schmidt et al., 2008, Werth et al., 2009, Werth et al., 2010 for WGHM and Sayed et al.,2008, for GLDAS (see P5 L16-20). Schmidt et al., 2008, compare different models (WGHM, GLDAS, LAD) with respect to phase shifts and come to the conclusion that these differences might point to systematic deficiencies in hydrological modelling. The motivation and the starting point for the development of the Cascaded storage approach was not only to clarify the problem of the phase shift, but to also describe the system behavior in a "Top-Down" approach by macroscopic parameters addressing the coupled / uncoupled storage compartment by their effect on the R-S relationship.

The thoughts and intention leading to its development are the following : - Summarizing all coupled storages contributing to mass and runoff and the uncoupled storages contributing to mass only allows to describe them in their accumulated effect on the runoff-storage relationship of the catchment without the necessity to address storages and flows in detail, while most hydrological models use a "Bottom Up" approach using spatial/temporal distribution of a number of parameters and driving forces. - A minimum number of input variables and optimization parameters shall be used to describe the system behavior without the necessity to use spatially / temporally distributed data and describe internal processes in detail. Catchment scale parameters shall be used instead to separate coupled and uncoupled storages (like MODIS snow coverage (Riegger and Tourian, 2014) ) - Recharge by atmospheric water balance via moisture flux divergence (P-ETa = divQ) or from catchment mass balance (P-ETa = dM/dt+R) on monthly scales is very convenient and quite accurate, however, it is limited to global scales (>200000km2), thus representing the ideal input for Top-Down approaches. - The linearity in R-S relationship allows for a piecewise analytical solution of the coupled balance equations for two different time constants and thus a mathematical description with no stability and accuracy criteria for the temporal discretization. For numerical solutions big differences in the time constants of the catchment and the river network lead to a high temporal discretization effort (see P9 L1-6). - The approach allows to determine river runoff from total drainable storage and vice versa directly from respective measurements once the time scales ïĄťˇ C and ïĄťˇ R are determined by an optimization (see P23 L26ff, codes and results implemented in the supplement). A related description is integrated in the manuscript now in detail. It has to be emphasized here, that this facilitates a purely data driven determination of runoff from drainable storage and vice versa with astonishing accuracy and thus permits a closure of data gaps without the necessity of additional model runs.

< Note that some of these models (such as WaterGap and PCR-GLOBWB) have groundwater parameterizations and are able to reproduce the amplitude and lags observed in GRACE (see e.g. Wada et al., 2012; Water Resour. Res., 48, W00L06).

-> According to Güntner et al., 2007, Schmidt et al., 2008, Werth et al., 2009, Werth et al., 2010, WGHM, which is related to WaterGap, actually shows a phase shift. However, I am not aware whether there are modifications between WaterGap and WGHM which enable a description of the R-S relationship without a phase shift. The model based total storage calculated by PCR-GLOBWB (Fig.1, Wada et al., 2012) seems to show some phase differences. As the phase shift is individual for each catchment according to the hydraulic and topographic conditions a direct comparison of the approaches or models for the Amazon catchment upstream Obidos would be necessary. Graphs for the Amazon are not contained in Fig.1 of Wada et al., 2012. A display of calculated total mass anomalies versus GRACE and of calculated versus measured runoff as shown below (and provided in the supplement) would allow to better recognize phase differences in a comparison. (see attached Fig.1 ) Thus, it would be quite enlightening for the discussion here if the reviewer would provide analogous graphs for the modeling results of WaterGap and PCR-GLOBWB related to the Amazon catchment. Furthermore a comparison of the optimization performance (according to Table 2) would be helpful.

< If the amplitude and phase shift between recharge and runoff are informative about storage and hence discharge, GRACE anomalies could be used to calibrate these models as well, with the added advantage that a) we do not need to assume linearity between storage and discharge; b) these models deal with temporarily unconnected storages as well.

-> Yes, hydrological models can be calibrated versus GRACE anomalies and runoff without any further assumptions on the R-S relationship, provided that the impact of river routing on river network mass and thus on total mass is described. (Any no zero temporal delay between catchment runoff into the river network and discharge at the catchment outlet leads to mass changes in the river network and thus in total storage (Eq.6).

a.) As investigations on the R-S relationship of global scale catchments (Riegger and

Tourian, 2014) have shown for fully humid and for boreal catchments (with temporally uncoupled storages), the relationship between runoff and coupled storage in fact is linear, leading to a Linear Time Invariant (LTI) system behavior. This need not be true for local or regional scale catchments, where thresholds might play a role. Any non linear R-S-relationship would lead to changes in the functional form of the resulting mass and runoff time series and not just to a pure phase shift. However, as the application to the Amazon catchment here (and to the boreal catchments in Riegger and Tourian, 2018) shows, the signal forms (Fig.10) are reproduced very well by the Cascaded storage approach and show a phase shift only (Fig.11) confirming the linear relationship. The mathematical framework for the Cascaded storage approach and the resulting consequences for the optimization properties are based on a linear storage. Thus the investigation of the optimization properties (based on a sinusoidal input for the synthetic case) with respect to the uniqueness and accuracy of the results cannot be transferred to non linear cases without further investigations. For catchments with non negligible river network storage the Cascaded storage approach is needed to describe the amplitude and phase shift between recharge and runoff with enhanced accuracy (see Column C in Table.2).

b.) The temporarily unconnected storages calculated in spatially / temporally distributed hydrological models on global scales can hardly be verified by ground based measurements. The Top-Down approach presented here does not describe unconnected storages in detail, but attempts to describe them by their impact on the R-S relationship (runoff independent contributions) based on a macroscopic, basin scale parameter, which is derived from additional information like remote sensing. This additional information is used for a separation of total storage into coupled and uncoupled storages (see P5L20-24) . For boreal catchments MODIS snow coverage serves as a separation parameter (Riegger and Tourian, 2018). For seasonally dry catchments the separation is still a major task as the uncoupled storages are dominated by open water bodies and soil moisture (see Outlook P24 L14-22).

< 3. Modelling the effect of the drainage networks as a linear storage-outflow relationship may be valid for the Amazon where during peaks the whole basin turns into a huge flooded area resembling a lake. But in many rivers of the world, e.g. the Danube, the Rhine, the Nile, water during high stages is confined in the channel or in narrow valleys and the lag between catchment discharge and discharge at the basin's outlet is more of a travel-time phenomenon then a storage attenuation phenomenon. In this case, a routing routine such as used in many global hydrological models would be more suitable, with the unknown parameters the channel and floodplain resistance parameters (e.g. Manning coefficient).

-> The Cascaded storage approach intends to describe global scale catchments considerably above the resolution limit ~200000km2 (like Amazon, Yenissei, Lena, Ob, Mackenzie, Yukon, Niger, Kongo, Mekong etc) in order to achieve reasonable accuracy for both, GRACE measurements and moisture flux divergence. The catchments mentioned (Danube, Rhine, Nile) are either too small or managed by hydraulic structures i.e. not draining without anthropogenic impacts. The approach here is a conceptual approach not claiming to describe internal processes like river routing. The hydraulic properties of the river network (topography, channel cross section and roughness, channel length and river gradient), which are not known very well on global scales, are summarized in this approach as one efficient hydraulic time constant ïĄ'R describing the overall river network dynamics and not the superposition of sub branches. Of course it would be interesting to compare the hydraulic time constants of river routing schemes with the one obtained by this approach. It will also be interesting (see Outlook P38 L23-26) to compare the river network mass (determined here in an independent way) versus the flood areas (GIEMS) or river / flood volumes from GIEMS and altimetry for other global scale catchments. This would provide insights into global scale river and flood hydraulics.

< 4. More generally: the approach seems to be valid in large humid basins, without cold-region processes, where all active stores are permanently connected to the

discharge mechanism, while routing is such that the drainage network can be represented by a storage-outflow relationship. This makes the applicability of the approach somewhat limited.

-> As mentioned in the "Introduction" (see P5 L20-24) in the "Conclusions and Discussion" (see P5 L20-24) a prerequisite for the approach is that the coupled and uncoupled storages can be separated by other means like remote sensing. For boreal regions with relatively homogeneous snow depth (opposite to mountainous regions) this can be done with the help of MODIS snow coverage (Riegger and Tourian, 2014). It has been shown that the snow covered parts represent the uncoupled, solid storage, while the open area represents the coupled, liquid part. This separation leads to a linear R-S relationship for the liquid part. Modelling of snow accumulation and melt over the snow covered parts and integrating the phase shift between runoff and total mass in the calculation scheme (Riegger and Tourian, 2014) leads to very reasonable results, thus confirming the benefits of a Top Down approach. The integration of the Cascaded storage approach into the calculation scheme for boreal catchments was beyond the scope of this publication and is the subject of present investigations. For seasonally dry catchments like Niger, Tocantins etc. the separation of the time dependent uncoupled storage compartments like soil or isolated surface water bodies remain a major task for remote sensing, i.e., for satellite soil moisture measurements and open water body altimetry (see Outlook (P24 L14-22).

< 5. The writing should definitely be improved. For instance, the abstract reads like an extended summary with an introduction and is too long and too specific. Also, the use of the English language should be checked by a native speaker. Suggestions for improvements and some other small remarks are given in the annotated manuscript attached.

-> According to the HESS guidelines for authors the abstract should comprise the motivation, an introduction into the method, a summary of the key points and directions of prospective research. This is actually the case. Nevertheless, the attempt was

made to shorten the text. The whole manuscript text has been checked by a native speaker and is revised accordingly. Changes are marked and integrated into the new version number 3.

Please also note the supplement to this comment:
https://www.hydrol-earth-syst-sci-discuss.net/hess-2018-38/hess-2018-38-AC1-supplement.zip
* * *
**Results of the Cascaded Storage approach compared to measurements**

**for the Amazon basin**

[Figure]

[Figure]

Calculated total drainable mass anomaly $dM_T$
versus GRACE signal

Calculated river runoff versus observed $R_o$
from HYBAM

**Fig. 1.** Scatter plots for total drainable storage volume and for river runoff

---

## Referee Comment (RC2) · A. Güntner (Referee) · 24 Jul 2018

This is an interesting and novel study on estimating the drainable water storage in large river basins based on observed discharge data and/or water storage anomalies from GRACE. The work develops an approach based on the linear storage concept to separate the total drainable storage volume of a river basin into two storage compartments, which are denominated the catchment storage and the river network storage. The manuscript comprehensively presents the methodological and mathematical concept of the approach and nicely illustrates the storage characteristics for the single and the two storage assumption in terms of signal dynamics, amplitudes and phases for

virtual experiments, including an assessment of uncertainties of the parameter estimation procedure. The concept is then applied to the real world example of the Amazon river basin, leading to interesting lumped results of basin storage and runoff dynamics. Nevertheless, I have some general doubts on the method and the way it is realized in this study:

1) The storage concept presented here, by looking at linear storages and their storage coefficients or 'time constants', takes a purely temporal perspective on catchment storage. It separates two storage compartments of different volume and drainage behaviour in time. While basically a viable approach, this is presumably not as straightforward as the manuscript implies when it comes to linking these quick and slow storages to a spatial (i.e. source area) perspective of storage and flow. For example, a quick runoff response may partly occur from the so-called catchment storage by, e.g., subsurface storm flow, whereas a slow response may also occur along the river network due to surface water-groundwater interactions or floodplain storage. Thus, I wonder whether the separation into a catchment and a river network storage as implied by the title can really be achieved by the method applied here, instead of a separation of a quick and a slow storage compartment.

2) As a prerequisite of the applicability of the approach, uncoupled storages (i.e. storage compartments that do not directly drain to the catchment outlet) need to be negligible or time invariant (page 23, lines 4-5). It is assumed that this condition is fulfilled in the study area Amazon basin (page 18). However, I doubt whether this assumption holds true. Given the strong seasonality of rainfall and evapotranspiration in large parts of the basin, there are substantial temporal variations of water storage in the unsaturated zone (e.g. Tomasella et al., 2008), including moisture states drier than field capacity of the soil, i.e., non-gravity-driven conditions. Such conditions correspond to storage variations in non-coupled storage compartments as defined for this study and were assumed to be negligible. This calls the approach into question.

3) The separation approach presented requires an estimate of recharge for the river
basin of interest. Three options are suggested (page 18). Following these suggestions, the input that should rather be called precipitation surplus, as commented by another referee, is not necessarily what it is claimed to be, i.e., it is not groundwater recharge or a similar flux term that contributes directly to a connected storage, but may at last partly go into an intermediate storage or it may experience travel times to the saturated zone given large groundwater depths in some parts of the Amazon basin. Thus, I wonder what the effect of this discrepancy between precipitation surplus and the required contribution to the connected storage is on the validity of the results and the values obtained here (e.g. of time constants).

4) The approach assumes a linear storage concept for representing the river network dynamics. As commented by another reviewer, this may not be adequate for several river basins. In particular, it does not apply to the Amazon basin given the particular dynamics of floodplains and inundation areas, and different gradients of large-scale water levels at the seasonal scales between the rising and falling limb of the annual flood wave.

5) It should be clarified to which extent the drainable storage values obtained for a particular river basin depend on the actual time period used in the analysis and on their particular observed storage amplitudes (which probably are smaller than what is physical reasonable and possible at the long-term), or whether they represent some fundamental catchment property.

6) While phase shifts between simulation results of hydrological models and GRACE storage variations exist as noted by the author, it is generally accepted that they can be attributed to model deficiencies in representing river flow routing or inundation dynamics, and the discrepancies may eventually be used to improve the model. In my view, parts of the manuscript that indicate that this study provides a new explanation for these phase shifts (e.g. page 5 last paragraph, page 23 line 11) may need to be re-written as I do not see this potential new contribution. In particular, given the lumped temporal nature of the approach presented here, the study does not contribute to a better understanding of reasons for phase shifts from a process-based perspective (page 5, line 21).

Minor comments:

7) As pointed out by another referee, the abstract needs to be shortened considerably, and the entire manuscript needs polishing of English language.

8) Hysteresis plots (Figure 1 and others): the direction of the hysteresis should be indicated.

9) Throughout the paper I suggest to use the term 'storage anomaly' instead of 'storage deviation', in line with what is usually used in literature.

10) page 11, line 21: 'signal amplitudes': first occurrence of this term which is not fully clear at this point. Explain here. Also the notion with a sigma sign for these signal amplitude ratios is somewhat confusing at the first instance.

11) page 12, line 17: '2D dependence': not clear.

12) page 19, line 2: 'monthly residuals': not clear at this point. (Only later it is explained that the de-seasonalized time series are meant here.)

13) page 20, figure caption 11: what does d in dMT mean? (if it is deviation, use mass anomaly)

14) page 23 and 24: equations may be skipped in the conclusions.

References:

Tomasella, J., Hodnett, M. G., Cuartas, L. A., Nobre, A. D., Waterloo, M. J., and Oliveira, S. M.: The water balance of an Amazonian micro-catchment: the effect of interannual variability of rainfall on hydrological behaviour, Hydr. Proc., 22, 2133-2147, 10.1002/hyp.6813, 2008.

38, 2018.

---

## Referee Comment (RC3) · Anonymous Referee #3 · 26 Jul 2018

**Title:** Quantification of Drainable Water Storage Volumes in Catchments and in River Networks on Global Scales using the GRACE and / or River Runoff

**Author:** Joannes Riegger

I carefully read the manuscript and comments from M. Bierkens and A. Gunter. They both raise important questions on the actual applicability of the proposed method. I found a few points in the manuscript that need more deepening. For that reason, I suggest that the paper could be considered for publication after review. Please find my comments below.

Major comments:

1. As pointed out by M. Bierkens, the abstract should be significantly shortened. I also recommend that the author avoid the use of concepts or terms in the abstract that are not properly explained. For example, it is not clear in the abstract alone, what the runoff-storage relationship (P.1, L. 17) is. It is also not clear what phase shift (P.1, L. 20) is being referred to. I also suggest that the Introduction should be rewritten, focusing on a clear statement of the issue the author is trying to address, a comprehensive literature review on what has been done before, and a simple description of how the problem will be tackled. Details on the technique should be reserved for the following sections.

2. My feeling is that there is a general lack of recent and appropriate literature in the field. For example, in the abstract, the author states: "*A possible reason for the observed phase shift might be found in the river network storage, which so far has not been addressed separately in the R-S relationships.*" Also, in the introduction: "*Very little attention is given so far to the storage volume of renewable water resources participating 5 in the dynamic water cycle driven by precipitation P, actual evapotranspiration ETa and river runoff R.*" Many modeling studies have been performed towards a better understanding of surface water storage (SWS) and dynamics. The impact of SWS on the terrestrial water storage variability is evaluated globally in Getirana et al. (2017a). In that study, the authors use Noah-MP, accounting for a detailed computation of the water and energy balances, including groundwater recharge, and an advanced river routing scheme, accounting for river and floodplain dynamics using the local inertia formulation.
In page 5, the author states: "*Even though global hydrological models comprise a number of storages like soil, surface water, groundwater etc. some of them show considerable phase shifts between the calculated and measured runoff and an underestimation of the signal amplitudes (Güntner et al., 2007, Chen et al., 2007, Schmidt et al., 2008, Werth et al., 2009, Werth et al., 2010).*" There are very well known reasons for these issues to happen, and the references used to support that statement are somehow outdated (8-11 years old). Recent developments on hydrological modeling, in particular, river routing schemes have successfully dealt with phase shifts and amplitude ratios in both Amazon and globally (Getirana et al., 2014, 2017b; Luo et al., 2017; Paiva et al., 2013; Yamazaki et al., 2011, 2012, 2014; Siqueira et al., 2018). I strongly suggest that the author better contextualize the study pointing out what the contribution is, considering what has already been done.

3. It is common sense to use the term runoff for the surface or total runoff generated by a land surface model, usually given by mm/d or mm/s, which is the rate of water flowing to the river network, while streamflow is used for the river discharge, usually in m3/s. The former is either simulated by LSMs or estimated from the spatial distribution of the latter, which can be observed at gauge stations.

Sometimes, in the text, I get confused with what the author is referring to. For example, in the abstract, the author refers to "observed runoff", while it should be "observed streamflow". I suggest that the author make a proper use of these terms and clarify when runoff and streamflow are used.

Minor comments:

1. In the paper, the application of the technique is limited to the Amazon, and I think that using the term "global scales" in the title is a bit of an overstatement. I suggest the removal of that term from the title.
2. P. 2, L. 1: Define Cascaded Storage approach
3. P. 2, L. 6: Define w.r.t.
4. P. 2, L. 31: "semi / arid" – Do you mean, semi-arid, or semi-arid and arid?
5. P.3, L19-20: "surface water, the river network and temporarily inundated areas" – what differentiates surface water from river network and temporarily inundated areas? It seems to me that the latter two are part of the former.
6. P. 6, L. 2: Define GIEMS
7. P. 6, L. 2-4: "Observations of inundated areas in river networks provided by the GIEMS project (Prigent et al, 2007, Paiva et al., 2013) indicate a considerable contribution of river network storage for the Amazon Catchment" – Getirana et al. (2012) provide the actual water storages in rivers and floodplains in the Amazon basin.
1. P. 6, L. 6: Paiva et al. (2013) is not a GIEMS reference.

I hope these comments will be useful for the preparation of an improved version of the paper.

Additional references

Getirana, A., Kumar, S., Girotto, M., Rodell, M., 2017a. Rivers and floodplains as key components of global terrestrial water storage variability. Geophysical Research Letters, 44. DOI: 10.1002/2017GL074684

Getirana, A., Peters-Lidard, C., Rodell, M., Bates, P.D., 2017b. Trade-off between cost and accuracy in large-scale surface water dynamic modeling. Water Resources Research. DOI: 10.1002/2017WR020519

Getirana, A., et al., 2014. Water balance in the Amazon basin from a land surface model ensemble. Journal of Hydrometeorology, 15, 2586-2614. DOI: 10.1175/JHM-D-14-0068.1 implementing the local inertial flow equation and a vector-based river network map, Water Resour. Res., 49, 7221–7235, doi:10.1002/wrcr.20552.

Luo, X., Li, H., Leung, L.R., Tesfa, T.K., Getirana, A., Papa, F., Hess, L.L., 2017. Modeling surface water dynamics in the Amazon Basin using MOSART-Inundation-v1.0: Impacts of

geomorphological parameters and river flow representation. Geoscientific Model Development, 10, 1233–1259. DOI: 10.5194/gmd-10-1233-2017

Papa, F., Frappart, F., Guntner, A., Prigent, C., Aires, F., Rossow, W.B., Getirana, A.C.V., Maurer, R., 2013. Surface Freshwater Storage and Variability in the Amazon basin from multi-satellite observations, 1993-2007. Journal of Geophysical Research, 118(21),11,951–11,965. DOI: 10.1002/2013JD020500

Siqueira, V. A., Paiva, R. C. D., Fleischmann, A. S., Fan, F. M., Ruhoff, A. L., Pontes, P. R. M., Paris, A., Calmant, S., and Collischonn, W.: Toward continental hydrologic–hydrodynamic modeling in South America, Hydrol. Earth Syst. Sci. Discuss., https://doi.org/10.5194/hess-2018-225, in review, 2018

Yamazaki, D., G. A. M. de Almeida, and P. D. Bates (2013), Improving computational efficiency in global river models by

Yamazaki, D., H. Lee, D. E. Alsdorf, E. Dutra, H. Kim, S. Kanae, and T. Oki (2012), Analysis of the water level dynamics simulated by a global river model: A case study in the Amazon River, Water Resour. Res., 48, W09508, doi:10.1029/2012WR011869.

Yamazaki, D., S. Kanae, H. Kim, and T. Oki (2011), A physically-based description of floodplain inundation dynamics in a global river routing model, Water Resour. Res., 47,W04501, doi:10.1029/2010WR009726.

---

## Author Comment (AC2) · 20 Sep 2018

<This is an interesting and novel study on estimating the drainable water storage in large river basins based on observed discharge data and/or water storage anomalies from GRACE. The work develops an approach based on the linear storage concept to separate the total drainable storage volume of a river basin into two storage compartments, which are denominated the catchment storage and the river network storage. The manuscript comprehensively presents the methodological and mathematical concept of the approach and nicely illustrates the storage characteristics for the single and the two storage assumption in terms of signal dynamics, amplitudes and phases for

virtual experiments, including an assessment of uncertainties of the parameter estimation procedure. The concept is then applied to the real world example of the Amazon river basin, leading to interesting lumped results of basin storage and runoff dynamics. Nevertheless, I have some general doubts on the method and the way it is realized in this study:

->Thanks to the referee for the effort. Below you find my respond to his comments in detail

< 1) The storage concept presented here, by looking at linear storages and their storage coefficients or 'time constants', takes a purely temporal perspective on catchment storage. It separates two storage compartments of different volume and drainage behaviour in time.

-> For clarification: Generally storage coefficients are defined as storage change versus changes in water head or pressure (P3 L12). Time constants are introduced by the exponential form of a flow from a linear storage if there is no input and are defined as proportionality coefficient between storage and runoff (Eq3).

< While basically a viable approach, this is presumably not as straight forward as the manuscript implies when it comes to linking these quick and slow storages to a spatial (i.e. source area) perspective of storage and flow. For example, a quick runoff response may partly occur from the so-called catchment storage by, e.g., sub- surface storm flow, whereas a slow response may also occur along the river network due to surface water-groundwater interactions or floodplain storage. Thus, I wonder whether the separation into a catchment and a river network storage as implied by the title can really be achieved by the method applied here, instead of a separation of a quick and a slow storage compartment.

-> As already mentioned in the "Introduction" flows from storages draining in parallel superpose, while storages in a sequence lead to a time lag or phase shift (P5 L22-24). Flows draining in parallel can be separated in the runoff curves directly by their

different response time constant, if there are distinct periods of negligible recharge like in seasonally dry regions (Niger, Tocantins, .etc.) long enough for a sufficient fit. (P4 L8-16). Averaged over the full Amazon catchment there are no dry periods (see below). While in different LSMs (Getirana, (2014) groundwater and surface (overland) flows are summed up as input into the river rooting procedure, they are conceptualized here as one flow component and thus one catchment storage (p6 L25-26). Actually, there is no other way to compare calculated streamflows with river discharge measurements, as they cannot be separated into contributions of different dynamic behavior for the full Amazon catchment.

-> The effect of the river network storage with an effective hydraulic time constant over the catchment does not correspond to the superposition of quick and slow flows, but instead leads to a phase shift (Eq27) between catchment storage (groundwater and surface flow storage) and the river network storage as it is in sequence. As it is shown in the "Parameter estimation" section the Cascaded storage approach is not limited to a faster response of the river network (ïĄťR < ïĄťC)) but also permits the description of river systems with a slow response (ïĄťR > ïĄťC)) as it may also occur "along the river network due to surface water-groundwater interactions or floodplain storage" of large river systems. However, as the phase shift (Eq27) is commutative (P16 L1-5) the assignment of the quick or the slow part of the storage to the catchment or river network mass is only possible with additional ground based or remote sensing information on the river network or floodplain extent (P17 L18-20).

-> Compared to inundated areas taken from Global Inundation Extent from Multi-Satellites GIEMS (see Fig.10) the calculated river network mass leads to correlation coefficients of 0.96 for the signal and 0.76 with respect to the mean seasonal cycle. This allows to assign the river network storage calculated here to the observed river network and flooded area volume. The average amplitude ratio MR/MT of the river network to total mass of 50% calculated here fits very well to the estimate of 50% by Papa, F., Frappart, F., Güntner, A., et al., (2013) and the ratio of 41% from Getirana et

al., (2017a).

-> In the light of the referee's own articles on surface water storage variability in large river basins (Papa, F., Güntner, A., Frappart, F., Prigent, C., et al., (2008)) and especially those for Amazon (Papa, F., Frappart, F., Güntner, A., Prigent, Aires, F., Getirana, A.C.V., (2013)), which come to the same results with different remote sensing methods, it is strange that the referee wonders whether this can be achieved by the proposed method without substantiating his wondering.

< 2) As a prerequisite of the applicability of the approach, uncoupled storages (i.e. storage compartments that do not directly drain to the catchment outlet) need to be negligible or time invariant (page 23, lines 4-5). It is assumed that this condition is fulfilled in the study area Amazon basin (page 18). However, I doubt whether this assumption holds true.

->The author is aware of the impact from time dependent uncoupled storages (Riegger and Tourian (2014)). Thus, additional prerequisites are formulated (P23 L 4-9). The prerequisites are extended for a general application of the scheme, such as : b) Separation of coupled and uncoupled storage compartments by conceptual approaches c) Full description of the hysteresis by quantified contributions of the coupled and uncoupled which are not mentioned here. The Amazon basin is chosen for first evaluations of the scheme as it fulfills prerequisite a.).

< Given the strong seasonality of rainfall and evapotranspiration in large parts of the basin, there are substantial temporal variations of water storage in the unsaturated zone (e.g. Tomasella et al., 2008), including moisture states drier than field capacity of the soil, i.e., non-gravity-driven conditions. Such conditions correspond to storage variations in non-coupled storage compartments as defined for this study and were assumed to be negligible. This calls the approach into question.

-> As data - provided in the supplement - indicate monthly recharge N = P-ET is positive for the full Amazon catchment upstream Obidos. This is also confirmed by the

study of Getirana et al. (2014) (Fig.5), where the monthly climatology of full Amazon is compared for 14 different global Land Surface Models.

-> In addition, any non negligible impact of time dependent uncoupled storages could be recognized in the R-S diagram and would also effect the scatter plots of simulated versus observed runoff or mass anomaly if the description of the uncoupled storage behavior were not sufficient.

-> For the full Amazon catchment this means that averaged over the full catchment area soil water content remains constant with permanent input, i.e. the uncoupled storage is time independent. This might not be the case for all sub catchments. In fact, dry out effects, i.e. mass changes without changes in runoff can be recognized in the R-S diagram of seasonally dry or monsoonal catchments with distinct wet and dry seasons like Niger etc (see P24 L17-20). According to above conditions b.) and c.) additional conceptual approaches or information from remote sensing are needed for the temporal description of the uncoupled storage. In Riegger and Tourian (2014) it has been shown for boreal regions that the uncoupled storage (in this case snow and ice) can be described satisfyingly by MODIS snow coverage. As emphasized in the outlook (P24 L17-20) for monsoonal regions the respective methods for the quantification of uncoupled storages by remote sensing (soil moisture and open water body altimetry from satellites) need to be developed.

-> It is one thing that the referee unfoundedly insinuates a time dependent uncoupled storage for the full Amazon catchment. It is however unintelligible that this could call the approach into question without explaining and supporting this judgement adequately in the light of the given prerequisites (P23 L 4-9).

< 3) The separation approach (i.e. Cascaded storage approach ??) presented requires an estimate of recharge for the river basin of interest. Three options are suggested (page 18). Following these suggestions, the input that should rather be called precipitation surplus, as commented by another referee, is not necessarily what it is claimed

to be, i.e., it is not groundwater recharge or a similar flux term that contributes directly to a connected storage, but may at last partly go into an intermediate storage or it may experience travel times to the saturated zone given large groundwater depths in some parts of the Amazon basin. Thus, I wonder what the effect of this discrepancy between precipitation surplus and the required contribution to the connected storage is on the validity of the results and the values obtained here (e.g. of time constants).

-> The expression recharge (Eq32-34) is used here to generally describe the lumped fluid input into a catchment received from either aggregated hydrometeorological data, the catchment's atmospheric water balance or the catchment's water balance using runoff and GRACE. It is not discriminating input leading to surface (overland) flow or groundwater flow (P6 L25-27), as for the Amazon catchment baseflow and surface (overland) flow components cannot be distinguished by observations. Thus, instead of using separate linear reservoirs for surface (overland) flow and groundwater flow as it is done in HyBam (Getirana et al. (2012), Eq.1) or in many LSMs (Getirana et al. (2014) only one linear reservoir is used here for simplicity for both flow contributions.

-> In WGHM (Döll et al., (2003) Eq.5) for example only one linear reservoir is used for groundwater flow yet none for surface flow with the consequence that the respective overland flow is routed to the river network without a delay. Generally, linear reservoirs for the different flow components are used in many hydrological models and LSMs in order to describe the dynamic system response including the time delay and the related storage volumes. The above mentioned models as well as WGHM do not describe flow in the unsaturated zone or other intermediate storage. The related transition times in this case are assumed to be negligible compared to the hydraulic time constant.

-> I wonder why the referee rises the complex question of travelling times in unsaturated zones and their impact on storage volume here as he - as a prominent user of WGHM - is certainly familiar with simplifications in LSMs and in the WGHMs calculation scheme.

< 4) The approach assumes a linear storage concept for representing the river network

dynamics. As commented by another reviewer, this may not be adequate for several river basins. In particular, it does not apply to the Amazon basin given the particular dynamics of floodplains and inundation areas, and different gradients of large-scale water levels at the seasonal scales between the rising and falling limb of the annual flood wave.

-> As already mentioned in my response to Mark Bierkens any non negligible, non linear contribution to the R-S-relationship would lead to changes in the functional form of the resulting mass and runoff time series and not to a phase shift only. Mathematically, only a linear R-S relationship in Eq7 can lead to a pure phase shift in the solution of Eq11 without any impact on the functional form. Data from the full Amazon catchment impose a behavior as a Linear Time Invariant (LTI) System (see Fig.1 for phase adapted mass and Riegger and Tourian, (2014)).

-> Complex river routing schemes such as HyBam used in the LSM comparison study of Getirana et al. (2014) come to similar results. With a Nash-Sutcliffe (NS) coefficient of 0.74 and a correlation (c) of 0.90 (with respect to the mean seasonal cycle) compared to an NS of 0.58 and a c of 0.84 for the best LSM (Getirana et al. (2014)) the Cascaded Storage approach outperforms the LSMs in combination with HyBam.
-> A possible explanation of this linear behavior between streamflow and storage for CATCHMENTS is that the river network system consists of many river branches, which interfere, and not of one branch or channel only. Hydraulically this might be understood, as flow in a river network with many contributing channels and branches behaves similarly to groundwater flow in a fractured system. The flow in a volume large enough to contain many contributing channels behaves like a porous continuum and can be described by Darcy's law -which is linear - instead of a discrete channel flow.

-> If the referee might have a closer look at the model he uses himself he will find out that in WGHM "the river itself is treated as a linear storage element similar to groundwater" (Döll et al., (2003), Eq.5)).

< 5) It should be clarified to which extent the drainable storage values obtained for a particular river basin depend on the actual time period used in the analysis and on their particular observed storage amplitudes (which probably are smaller than what is physical reasonable and possible at the long-term), or whether they represent some fundamental catchment property.

-> The time constants adjusted within the optimization periods might slightly change with the length of the period, yet generally are considered as a kind of fundamental catchment property as long as there are no significant changes in land surface properties or river hydraulics and as long as no anthropogenic impacts occur.

-> Runoff statistics for Amazon upstream Obidos deliver for GRDC measurements :

- R = 42.3 – 170.5 , average R = 99.5 stdev 28.8 [mm/mo] for 1980-2008

- R = 45.5 – 140.5 , average R = 96.2 stdev 26.2 f[mm/mo] or 2003-2008

and for Hybam measurements :

- R = 45.5 – 155.5 , average R = 96.2 stdev 26.2 [mm/mo] for 2003-2008 used in the study here

-> This means that – apart from some high discharge events before 2003 - the range covered by the modelling period corresponds to the long term statistics.

-> As the observed runoff does not cover a range from zero to the observed minimum, a superposition of a storage with a much longer time constant and thus leading to a very small contribution to runoff might not be visible at present. An additional storage release could come from a deep confined aquifer underneath the unconfined aquifer close to the surface. Most of the hydrological and LS models (like WGHM (Döll et al., (2003) amongst others) do not consider more than one groundwater storage.

-> However, in this study no deviation from the linear behavior caused by a contribution of such an aquifer can be observed at the present runoff range.  -> Generally, the

possible impact of such a storage means that the drainable storage determined from the mass and runoff range in this study is a lower limit of possible drainable storages.

-> An extrapolation to a runoff range beyond the maximum will have to face the same challenge as other hydrological or LS models such as WGHM (Döll et al., (2003), which use a linear storage for the river network.

-> I am wondering why the referee creates the impression that the problem of an extrapolation beyond the parameter range used for optimization is a specific problem of the Cascaded storage approach but letting unmentioned that all hydrological or LS models using a linear storage for the river system – including WGHM - have to face the same problem.

< 6) While phase shifts between simulation results of hydrological models and GRACE storage variations exist as noted by the author, it is generally accepted that they can be attributed to model deficiencies in representing river flow routing or inundation dynamics, and the discrepancies may eventually be used to improve the model.

-> In his paper (Schmidt, R., Petrovic, S., Güntner, A., (2008)) the referee states that the phase shift "points to systematic deficiencies in hydrological modeling. For example, water storage in surface water bodies will cause a delay of freshwater runoff from continental areas. However, processes of runoff routing in the river network and lake/wetland water retension are not taken into account by hydrological model versions used in this study, except for WGHM". However, he does not mention how model deficiencies are removed in WGHM. In Werth S., Güntner, A. (2010) he reports that "WGHM still tended to underestimate seasonal TWS variations and phase shifts appeared".

-> Thus, it would be quite illuminating if he would provide recent simulation results from WGHM for runoff as well as total and river network mass for the Amazon basin upstream Obidos and sort their performance into the comparison study of the LSMs by Getirana et al., (2014).

< In my view, parts of the manuscript that indicate that this study provides a new explanation for these phase shifts (e.g. page 5 last paragraph, page 23 line 11) may need to be re-written as I do not see this potential new contribution.

-> According to science theory there are many ways to Rome. Therefore, a hypothesis or approach is accepted if it gives reasonable physical and mathematical explanations for an observed effect, if it describes it sufficiently and if it does not lead to contradictions. This includes approaches on different temporal and spatial scales.

-> In their spatially distributed approach combining 14 different LSMs and the river routing scheme of HyMap Getirana et al., (2014) show that implementing an appropriate routing schemes permits an appropriate description of the TWS amplitude. In their study of Rivers and Flood plains storage Variability Getirana et al. (2017a) they state that "Adding SWS (corresponding to river network storage) and LWS (corresponding to catchment storage) improves the phase agreement with GRACE based observations". They emphasize that "SWS is a major component of TWS variability in the tropics. . ., where that storage component has been neglected in a series of previous hydrological studies".

-> In the study here a different approach is chosen for a different purpose. Based on the reported LTI behaviour of coupled storage and runoff for the entire catchment the absolute drainable storage volume shall be determined. This goes beyond the description of storage variations. Thus, for a lumped description of catchments by their mass balance an appropriate concept must be found in order to describe the effect of the phase shift by a physical concept. (In previous studies (Riegger and Tourian, (2014) the phase shift had been integrated into the numerical calculation scheme without the necessity of a detailed understanding (P5 L 21), yet leading to a description of the system behavior with high accuracy (NS=0.6 and correlation=0.85 for full Amazon w.r.t. mean seasonal cycle)).

-> In order to account for the phase shift the river network is conceptualized as one

(lumped) storage with an effective (lumped) time constant in sequence to the catchment storage comprising surface (for overland flow) and groundwater storage. This permits a piecewise analytical description of the coupled system without the disadvantages of numerical calculations. This concept and its realization is a new contribution to global scale modeling.

-> The Cascaded storage approach, in which the system behavior of the land surface AND the river network is described by the two time constants $t_c$, $t_R$ exclusively, leads to the same conclusions and to analogous results as Getirana et al., (2014), yet from a different perspective.

< In particular, given the lumped temporal nature of the approach presented here, the study does not contribute to a better understanding of reasons for phase shifts from a process-based perspective (page 5, line 21).

-> As the title indicates the main purpose of this paper is to determine the absolute drainable storage volume of catchments based directly on observations from GRACE and runoff. It was never intended to describe the system from a process based perspective.

-> In this paper the physical effects of a storage cascade on the phasing and signal amplitude are investigated extensively in order to clarify, if an optimization versus GRACE anomalies or measured runoff leads to unique results in absolute storage volume. Its application to the full Amazon basin provides a very accurate description of phase and amplitude for runoff and mass and in addition an independent, quite accurate description of the river network volume.

===> Looking at the structure of his comments it is obvious that the referee puts approaches and concepts into question without substantiating his wondering. The referee expresses his doubts, however, neither explains and supports his judgements adequately nor reveals possible contradictions or inconsistencies. He doubts and wonders about approaches which are generally accepted in other modeling concepts and which

are even used by himself applying WGHM. So I think, it is up to the referee to prove how a wrong approach with wrong assumptions can lead to a model performance as presented here. All data and calculations are provided in the supplement.

Minor Comments

The referees minor comments will be considered in the revision of the paper.

References: Döll, P.,F. Kasper, and B. Lehner A global hydrological model for deriving water availability indicators: model tuning and validation, J. Hydrol.,270,105-134, 2003 Getirana, A., Kumar, S., Girotto, M., Rodell, M. Rivers and floodplains as key components of global terrestrial water storage variability. Geophysical Research Letters, 44. DOI: 10.1002/2017GL074684., 2017a Getirana, A., et al. Water balance in the Amazon basin from a land surface model ensemble. Journal of Hydrometeorology, 15, 2586-2614. DOI: 10.1175/JHM-D-14-0068.1 implementing the local inertial flow equation and a vector-based river network map, Water Resour. Res., 49, 7221–7235, doi:10.1002/wrcr.20552, 2014 Papa, F., Güntner, A., Frappart, F., Prigent, C., Rossow, W.B.: Variations of surface water extent and water storage in large river basins: A comparison of different global sources, Geophysical Research Letters, 35, doi: 10.1029/2008GL033857, 2008 Papa, F., Frappart, F., Güntner, A., Prigent, Aires, F., Getirana, A.C.V., Maurer, R.: Surface freshwater storage and variability in the Amazon basin from multi-satellite observation, 1993-2007, Journal of Geophysical Research: Atmospheres, 118, doi: 10.1002/2013JD020500, 2013 Schmidt, R., Petrovic, S., Güntner, A., Barthelmes, F., Wünsch, J., and Kusche, J.: Periodic components of water storage changes from GRACE and global hydrology models. J. Geophys. Res., 113, B08419, oi:10.1029/2007JB005363, 2008 Werth, S., Güntner, A. Calibration analysis for water storage variability of the global hydrological model WGHM. Hydrol. Earth Syst. Sci., 14, 59–78, 2010

---

## Author Comment (AC3) · 20 Sep 2018

< I carefully read the manuscript and comments from M. Bierkens and A. Gunter. They both raise important questions on the actual applicability of the proposed method. I found a few points in the manuscript that need more deepening. For that reason, I suggest that the paper could be considered for publication after review. Please find my comments below.

-> I would like to thank the referee very much for his helpful comments especially with respect to very important references.

Below you find my respond to his comments in detail.

< 1. As pointed out by M. Bierkens, the abstract should be significantly shortened. I also recommend that the author avoid the use of concepts or terms in the abstract that are not properly explained. For example, it is not clear in the abstract alone, what the runoff- storage relationship (P.1, L. 17) is. It is also not clear what phase shift (P.1, L. 20) is being referred to. I also suggest that the Introduction should be rewritten, focusing on a clear statement of the issue the author is trying to address, a comprehensive literature review on what has been done before, and a simple description of how the problem will be tackled. Details on the technique should be reserved for the following sections.

-> The abstract will be revised according to the recommendations.

< 2. My feeling is that there is a general lack of recent and appropriate literature in the field. For example, in the abstract, the author states: "A possible reason for the observed phase shift might be found in the river network storage, which so far has not been addressed separately in the R-S relationships." Also, in the introduction: "Very little attention is given so far to the storage volume of renewable water resources partic- ipating 5 in the dynamic water cycle driven by precipitation P, actual evapotranspiration ETa and river runoff R." Many modeling studies have been performed towards a better understanding of surface water storage (SWS) and dynamics. The impact of SWS on the terrestrial water storage variability is evaluated globally in Getirana et al. (2017a). In that study, the authors use Noah-MP, accounting for a detailed computation of the water and energy balances, including groundwater recharge, and an advanced river routing scheme, accounting for river and floodplain dynamics using the local inertia formulation.

-> I am very grateful for the literature links to recent investigations. Obviously my literature alarm did not work very well.

-> It is very interesting to see that complementary work has been done in parallel with

very similar results and conclusions using totally different approaches namely high performance distributed models on the one hand and a simple conceptual, top down model on the other hand.

-> The publications listed are very helpful to position the Cascaded approach presented here in the context of river routing investigations and Surface Water Storage. Especially the water budget investigations for 14 different LSMs (Getirana et al. (2014)) are very helpful as they allow to sort in the results of the Cascaded Approach into those of the LSMs (Getirana et al. (2014), Fig. 14). With a Nash-Sutcliffe (NS) coefficient of 0.74 and a correlation (c) of 0.90 (with respect to the mean seasonal cycle) compared to an NS of 0.58 and a c of 0.84 for the best LSM the Cascade Storage approach outperforms the LSMs. Yet, this is mainly seen as the result of the quality of recharge data taken from the water balance using GRACE and river runoff as the use of moisture flux divergence for this purpose leads to much worse results. This limits the approach to a lumped description of basins on global scales due to the resolution limits of GRACE. Yet, with improvements in the spatial resolution of gravity satellites the number of catchments which can be described by this approach will tremendously increase (P24 L26-28).

-> However, the Cascaded Storage approach was never intended to compete with the LSMs providing spatial distributions of water budget variables, but instead to enable a purely data driven determination (and forecast) of river discharge from GRACE as well as absolute, drainable storage volumes for catchments and river networks in a simple, lumped approach directly from satellite data (i.e. in general from GRACE and additional remote sensing data). No detailed information on vegetation, soil etc., complex flow processes nor detailed hydraulic information for river routing like roughness, cross section, gradient or backwater effects is needed. Of course, the simplicity and accuracy of this approach is payed by the lack of the spatial information within the catchment. However, spatial distributions within catchments are difficult to be evaluated locally anyway.

< In page 5, the author states: "Even though global hydrological models comprise a number of storages like soil, surface water, groundwater etc. some of them show considerable phase shifts between the calculated and measured runoff and an under-estimation of the signal amplitudes (Güntner et al., 2007, Chen et al., 2007, Schmidt et al., 2008, Werth et al., 2009, Werth et al., 2010)." There are very well known reasons for these issues to happen, and the references used to support that statement are somehow outdated (8-11 years old). Recent developments on hydrological modeling, in particular, river routing schemes have successfully dealt with phase shifts and amplitude ratios in both Amazon and globally (Getirana et al., 2014, 2017b; Luo et al., 2017; Paiva et al., 2013; Yamazaki et al., 2011, 2012, 2014; Siqueira et al., 2018).

-> I am very grateful for these references !

-> It is interesting to see that totally different approaches, a sophisticated bottom up and a simple conceptual top-down approach lead to similar results. The average river network storage contribution of 50% calculated here is equal to the 50% by Papa et al. (2013) and close to the 41% by Getirana et al. (2017a).

-> Getirana et al. (2017a) also confirm that the large Surface Water Storage SWS in the Amazon basin increases the simulated TWS toward a better match with GRACE. It also confirms that adding SWS improves the phase agreement with GRACE. They come to the conclusion that SWS (called river network storage here) is a major component of Total Water Storage TWS and they emphasize the importance of integrating adequate river routing schemes and the consideration of SWS when composing or decomposing TWS.

-> This consistency with the Cascaded storage approach institutes new possibilities for investigations on the hydraulic time constant of river networks and on the relationship between flood areas, volumes, river runoff and calculated river network mass in general. This possibly provides deeper insights into river hydraulics i.e. routing schemes and the mass-, area-, and level- relationships of flooded areas (P24 L23-25).

< I strongly suggest that the author better contextualize the study pointing out what the contribution is, considering what has already been done.

-> The recommended references certainly help to further sharpen the intention of the Cascaded storage approach and its benefits in the context of parallel investigations. All aspects that turned up with these references will be integrated into the revise publication.

< 3. It is common sense to use the term runoff for the surface or total runoff generated by a land surface model, usually given by mm/d or mm/s, which is the rate of water flowing to the river network, while streamflow is used for the river discharge, usually in m3/s. The former is either simulated by LSMs or estimated from the spatial distribution of the latter, which can be observed at gauge stations. Sometimes, in the text, I get confused with what the author is referring to. For example, in the abstract, the author refers to "observed runoff", while it should be "observed streamflow". I suggest that the author make a proper use of these terms and clarify when runoff and streamflow are used.

-> The expression "Runoff R" (here in mm/month) is used here as general expression for the drainage density i.e. the drainage rate per area for the respective catchment as it is needed for a comparison with mass density (Eq2) and for the mass balance equation Eq 4. "River Discharge" denoted as Q (Eq1) is used for the streamflow (i.e. in m3/s) measured at gauging stations. "River runoff" is thus given by river discharge divided by the catchment area.

-> Runoff is not generally used as a synonym for overland flow or flow in the river network, yet is specified by an index C for the catchment and R for the river network system. Catchment runoff in this paper conceptionally comprises all contributions to catchment drainage whether this is overland flow or groundwater flow, as due to the climatic and hydraulic conditions of the full Amazon basin upstream Obidos these flows cannot be separated in the measurements.

-> This would be different for boreal or seasonally dry regions (P16 L22-24 ) where contributions of overland and groundwater flow can be distinguished by their dynamic response. Disregarding the phase shift catchment and river runoff are of the same mean values.

-> The manuscript will be checked for consistency and revised for a clear description.

Minor comments

< 1. In the paper, the application of the technique is limited to the Amazon, and I think that using the term "global scales" in the title is a bit of an overstatement. I suggest the removal of that term from the title.

-> The expression "Global scales" in the title is used to indicate that the approach is limited to catchment areas well above 200000 km2 due to the spatial resolution of GRACE and to the structure length of moisture flux divergence.

-> The approach is neither limited to the Amazon basin nor to fully humid catchments. This has already been shown by Riegger & Tourian 2014 for boreal regions, for which the uncoupled storage was quantified by means of remote sensing (MODIS snow coverage). As already mentioned (P16 L25-30) the challenge for seasonally dry or monsoonal catchments, where soil moisture plays a major role in the annual cycle, is to quantify the uncoupled storage compartments by other means of remote sensing like satellite soil moisture, water level altimetry etc.

-> Thus, even though the application of the scheme is evaluated on the Amazon basin for a simplified start (no consideration of time dependent uncoupled storage), it is not limited to this specific catchment. The approach – as formulated here - can be applied to all fully humid tropical catchments.

-> Applications with time dependent uncoupled storage are more complex and need the integration of remote sensing data as shown in Riegger and Tourian, (2014) for boreal catchments (P5 L7). For this case additional lumped information (snow cov-

erage from MODIS) can be used for a separation of coupled and uncoupled snow / ice storages leading to a reasonable accuracy of total mass estimations (NS=0.37 and correlation=0.69 for Lena w.r.t. mean seasonal cycle)), which is in the range of LSM performances. The re-formulation with the Cascaded storage approach is under work. Further developments and investigations for monsoonal catchments as mentioned in the outlook are proposed for funding.

-> Of course the quantification of uncoupled storage compartments by remote sensing is a real challenge at the moment, yet the perspectives to apply this method on a global coverage are quite promising.

< 2. "Cascaded" will be defined

< 3. w.r.t. means with respect to

< 4. "semi / arid" – Do you mean, semi-arid, or semi-arid and arid? -> means semi-arid and arid

< 5. "Surface water, the river network and temporarily inundated areas" – what differentiates surface water from river network and temporarily inundated areas? It seems to me that the latter two are part of the former.

-> Surface water in this context means the storage related to overland flow not including the river network and inundated areas. Isolated surface water bodies are not considered here. Inundated areas which are not isolated are assigned to the river network system. So to be clear, overland flow storages should be mentioned here instead of possibly misleading surface storages.

-> The text will be checked and revised for a clear description.

< 6. GIEMS means "Global Inundation Extent from Multi-Satellites"

< 7. "Observations of inundated areas in river networks provided by the GIEMS project (Prigent et al, 2007, Paiva et al., 2013) indicate a considerable contribution of river

network storage for the Amazon Catchment" – Getirana et al. (2012) provide the actual water storages in rivers and floodplains in the Amazon basin.

-> Thanks ! This was a very useful hint.

< 8. Paiva et al. (2013) is not a GIEMS reference

-> Sorry, that might have been mixed up in the text. Paiva et al., (2013) describe a quite sophisticated modeling approach for the Amazon basin including the hydrodynamic modeling of backwater effects. It is interesting that they come to an average surface water contribution (corresponding to river network storage here) of 56%, which is close to the value determined here.

---

## Author Response (AR1)

**Editor Decision: Reconsider after major revisions (further review by editor and referees)** (10 Oct 2018) by Stan Schymanski Comments to the Author:

**Thank you for the detailed responses to the referee comments. All referees recommend major revisions before the manuscript can be reconsidered for publication, and I will follow this recommendation.**

Thanks to the editor for the review. The comments of referees Mark Bierkens and Anonymous 3 are really helpful and certainly help to improve the paper. I will implement all issues addressed in the comments in the revised version.

While following up on the referee comments and your responses, I stumbled over a very recent paper by one of your former co-authors:

Tourian, M. J., Reager, J. T. and Sneeuw, N.: The Total Drainable Water Storage of the Amazon River Basin: A First Estimate Using GRACE, Water Resources Research, 54(5), 3290–3312, doi:10.1029/2017WR021674, 2018. This a very relevant paper that uses the same data and gets to very similar results in terms of estimating the total drainable water storage of the Amazon basin. The paper goes even further and analyses water table data and subcatchments within the Amazon basin. I am aware that Tourian et al. (2018) use the data in a different way to your work and that it was published after submission of your manuscript, but in order for the advantages of your approach to be understood, the manuscript will need to be put into relation to Tourian et al. (2018) and other recent papers, as pointed out by the referees.

The editor states that this paper goes even further than my manuscript and that it describes analyses of Amazon sub-catchments.

There are two parts in the study of Tourian et al. (2018). The first is related to the determination of drainable storage making use of the R-S relationship, the second part determines the river network volume and compares it to GIEMS observations. Relevant for the discussion here is mainly the first part. This describes nothing else but the application of the direct approach of Riegger and Tourian (2014) to the sub-catchments of the Amazon with a different phase shifting scheme than the one used in the original. However, the study does not consider the prerequisites of the approach described by Riegger and Tourian (2014), namely, that for a determination of the time constant from the R-S relationship the hysteresis must be fully described by a phase shift. This condition is fulfilled, if after the application of the phase shift, no systematic deviations from a linear relationship are observed. The consequence is, that before the scheme can be applied, it must be made sure that there is no impact from uncoupled storages and that there is no other impact on the hysteresis other than a pure phase shift.

In Tourian et al (2014), however, the scheme is applied without addressing the individual R-S diagrams of the Amazon sub-catchments in detail.

My own investigations of the Amazon sub-catchments reveal, that some of the sub-catchments (DBID 501) fulfill the prerequisites in the same way as the full Amazon catchment (DBID 295). The counter clockwise hysteresis corresponds to a positive phase shift and is explainable by a time lag (Fig 1)

Fig.1 I: mean monthly values of observed runoff versus GRACE mass anomaly dM for catchments with counterclockwise hysteresis.caused by time lag

Other catchments (DBID 504, 506, Fig.2) show a totally different behavior with a clockwise hysteresis (as other seasonally dry catchments like Niger etc (Riegger and Tourian (2014)) and a form which does not correspond to a time lag.

---

## Author Response (AR3)

According to Theresa Blume's (Executive Editor) comments and advice the manuscript has been revised with regard to the following issues :

- „shallow groundwater" at a prominent position
  - ➔ P8, L9-15
- Explanation why the water storage in the unsaturated zone is irrelevant for the Amazon
  - ➔ P8, L21-25, P21, L28-30-P22, L1-2
- „Conzept of Catchment Runoff „
  - ➔ P8, L9-15
- Citation of the 2nd proposed paper
  - ➔ P4, L10-11, P32 L5-7
- Separation of Conclusion and Discussion. „
  - ➔ P27 L4 ff, p28, L11
- Revision of Sub- and superskripts in the text
  - ➔ replaced

[revised manuscript text omitted]

Table.2: The statistical characteristics are listed for calculated river runoff $R_R^R$, total mass $M_T^T$, basin mass $M_C^C$ and river network mass $M_R^
[revised manuscript text omitted]

---

## Author Response (AR4)

According to the Editors comments and advice the manuscript has been revised with regard to the following issues :

- Eqs 1 and 2: Thinking about the comment by Referee #4 that "an introductory discussion of how basin storage can drive river flow" is missing, I realised that Eq. 1 comes a bit out of the blue.
→ Text revised P2L30 – P3 L10

- P4L21: This should be tau=M/R, for consistency with Eq. 2.
→ revised

- Eqs. 4-7: Please follow the HESS convention where each equation stands on its own line with its own label, e.g.:
"The total system behaviour is described by two balance equations, one for catchment storage (Eq. 4) and one for river storage (Eq. 6):"
followed by Eqs. 4-7, where the word "with" before 5 and 7 would be on a separate line before the equation.
→ revised

- Eq. 8: Why do you call the "time dependent recharge" N(w), not N(t), as in Eq. 9?
→ revised

- P28L11-: In the track-changes file, this is labelled "Discussion", but the discussion has to come before the conclusions. Please fix the formatting of P28L11 and move this Discussion section before the Conclusions.
I would also remove the first sentence of the Discussion: "Comparing global hydrological or land surface models with the presented Cascaded storage approach it can be summarized:"
→ revised

The manuscript still contains a large number of language-related and typographical errors and formatting that is not consistent with HESS guidelines
(e.g. in-text citations, where the year should be in brackets), but I believe that this will be fixed during the copy-editing process.
→ revised